# Stochastic gene expression in *Arabidopsis thaliana*

Ilka Schultheiß Araújo[1], Jessica Magdalena Pietsch[1], Emma Mathilde Keizer[2], Bettina Greese[3], Rachappa Balkunde 🔬 [1], Christian Fleck[2] & Martin Hülskamp[1]

Although plant development is highly reproducible, some stochasticity exists. This developmental stochasticity may be caused by noisy gene expression. Here we analyze the fluctuation of protein expression in *Arabidopsis thaliana*. Using the photoconvertible KikGR marker, we show that the protein expressions of individual cells fluctuate over time. A dual reporter system was used to study extrinsic and intrinsic noise of marker gene expression. We report that extrinsic noise is higher than intrinsic noise and that extrinsic noise in stomata is clearly lower in comparison to several other tissues/cell types. Finally, we show that cells are coupled with respect to stochastic protein expression in young leaves, hypocotyls and roots but not in mature leaves. Our data indicate that stochasticity of gene expression can vary between tissues/cell types and that it can be coupled in a non-cell-autonomous manner.

[1] Botanical Institute, Biocenter, Cologne University, 50674 Cologne, Germany. [2] Laboratory for Systems and Synthetic Biology, Wageningen University, 6703 HB Wageningen, The Netherlands. [3] Computational Biology and Biological Physics, Faculty for Theoretical Physics and Astronomy, Lund University, 223 62 Lund, Sweden. Correspondence and requests for materials should be addressed to C.F. (email: christian.fleck@wur.nl) or to M.Hül. (email: martin.huelskamp@uni-koeln.de)

Plant development is governed by regulatory mechanisms that lead to the formation of specialized cell types and tissues in a well-organized manner. At the cellular and molecular level, however, a surprisingly high degree of stochasticity is observed[1]. One way to look at stochasticity is that it may be a problem to establish regularity. On the other hand, stochasticity might be important to break the homogeneity, which is necessary for correct pattern formation[2,3].

In plants, stochasticity during development is best described for leaf growth. Here no correlation between growth rates and cell sizes, nuclear sizes and anisotropy was found[4]. Similarly, the length of the cell cycle and the time point at which cells switch to endoreduplication was found to be stochastic in sepals[5]. Recently, it was demonstrated that fluctuations of the transcription factor ATML1 initiate the spatial distribution of giant cells in sepals[6]. The characteristics and basis of stochastic gene expression was analyzed in various organisms including bacteria, yeast, mammalian cell cultures, *Dictyostelium discoideum, Mus musculus* and *Drosophila melanogaster*[7–15]. The overall noise of gene expression in a given cell can be divided into two components[8]. Extrinsic noise equally affects the expression of all genes in a cell, for example, because of differences in the number of RNA polymerases or ribosomes between cells. Intrinsic noise is due to the inherent stochasticity of molecular processes influencing transcription and translation. As a consequence, the expression of individual genes fluctuates over time.

In this work we analyze the noisiness of gene expression in *Arabidopsis thaliana* with emphasis on two questions: First, is intrinsic and extrinsic noise different in different tissues or cell types? It might be expected that stochasticity changes during cell differentiation or endoreduplication. Endoreduplication leads to higher copy numbers of genomes, which could balance the fluctuation of individual gene copies and a reduction of intrinsic noise. Second, is stochasticity of gene expression coupled between cells in a tissue? This could be the case because cellular conditions are inherited during cell divisions or because plant cells are well connected with each other through plasmodesmata[16] such that they could cross regulate and balance each others transcription.

We demonstrate that gene expression fluctuatuates over time. In addition, we show that extrinsic noise is higher than intrinsic noise and that extrinsic noise in stomata is lower than in other tissues/cell types. Our spatial analysis of stochastic gene expression revealed coupling between cells in some but not all tissues.

## Results

**Temporal analysis of fluctuations.** Fluctuations of gene expression over time have been successfully measured in single-cell

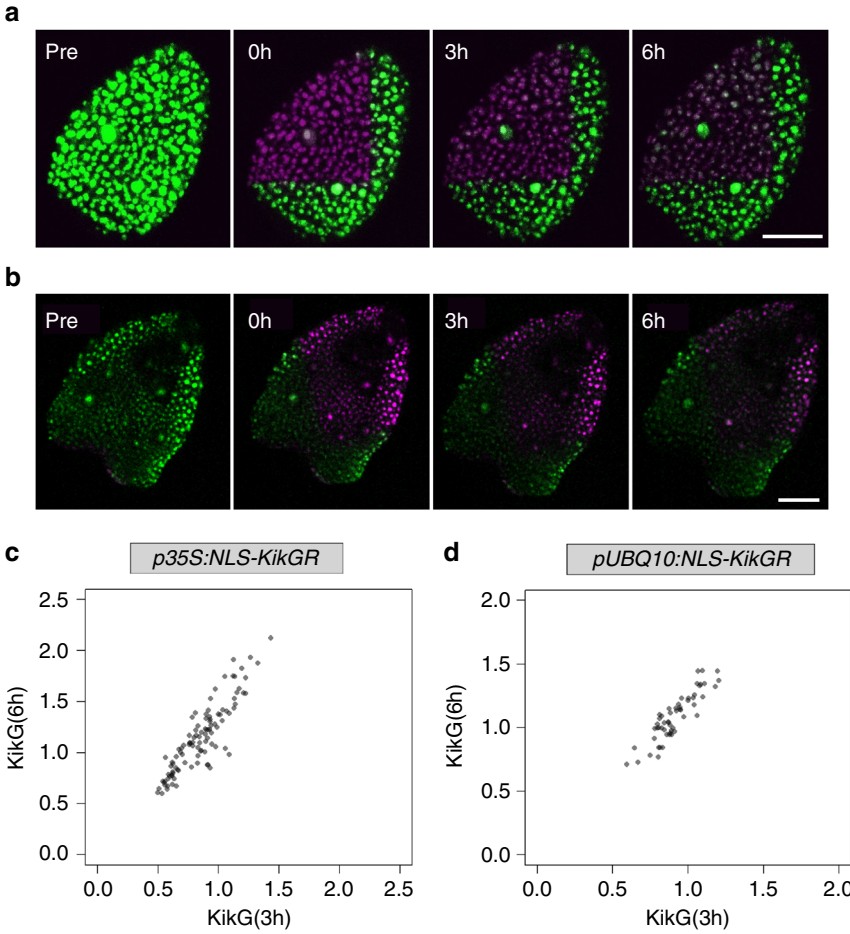

**Fig. 1** Temporal analysis of fluctuation in *p35S:NLS-KikGR* and *pUBQ10:NLS-KikGR* lines. **a** Confocal laser scanning microscopy (CLSM) images of *p35S:NLS-KikGR* before (pre) and after conversion (0 h, 3 h and 6 h). **b** CLSM images of *pUBQ10:NLS-KikGR* before (pre) and after conversion (0 h, 3 h and 6 h). Scale bar: 50 μm. **c** Scatter plot of *p35S:NLS-KikG* expressing cells (n = 103) obtained from one representative leaf. The normalized mean fluorescence intensity of the cells at 3 h is plotted against the normalized mean fluorescence intensity of the cells at 6 h. Data points are shown in grey, overlapping data points appear black. **d** Scatter plot of *pUBQ10:NLS-KikG* expressing cells (n = 55) obtained from one representative leaf

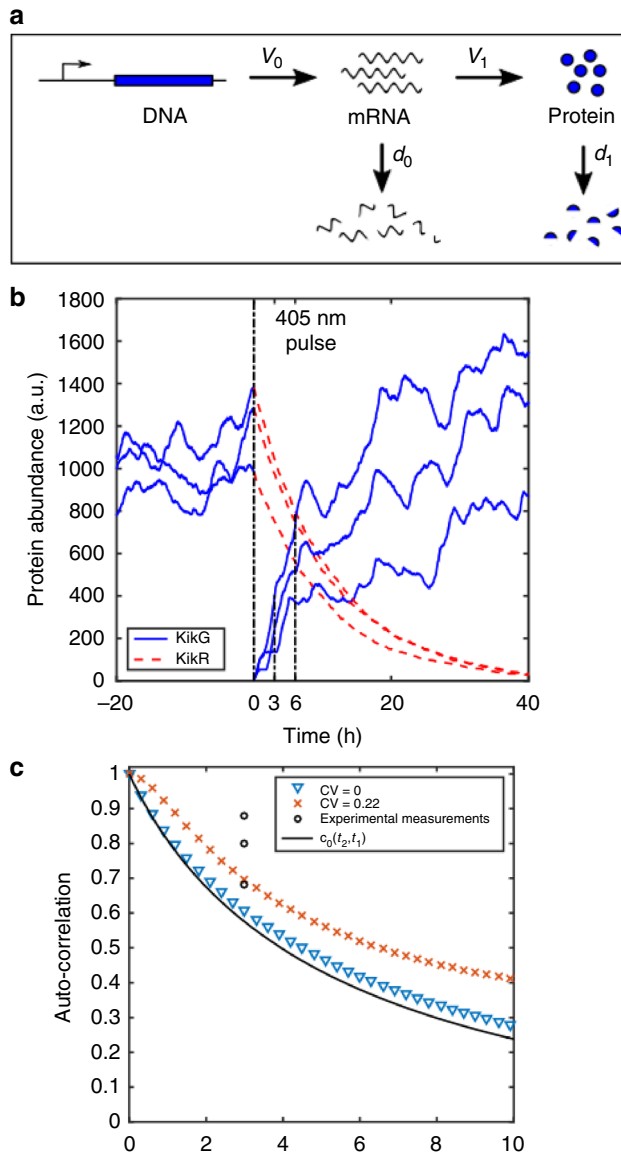

**Fig. 2** Theoretical analysis of fluctuations. **a** Schematic illustration of the two-stage stochastic gene expression model. $v$ = production rate, $d$ = degradation rate. (**b**) Modelling of three stochastic realizations of the KikGR reporter. After a 405 nm pulse the green fluorescence-emitting KikG is transformed into red fluorescence-emitting KikR. The auto-correlation between KikG at 3 h and 6 h is calculated. Parameters are: $\nu_0 = 2.25$ h$^{-1}$, $d_0 = 1.125$ h$^{-1}$, $d_1 = 0.09$ h$^{-1}$ and $\nu_1 = 41.825$ h$^{-1}$, $\nu_1 = 48.506$ h$^{-1}$, $\nu_1 = 46.069$ h$^{-1}$ for the three different trajectories. (**c**) Modelling of the non-stationary auto-correlation of the two-stage gene expression model in presence of extrinsic noise (crosses and triangles) as calculated from stochastic simulation of the KikGR reporter from $10^5$ trajectories (Supplementary Note 1) and the theoretical non-stationary auto-correlation of a birth-death process $c_0(t_2, t_1)$ (black solid line) as a lower bound of the non-stationary auto-correlation (Supplementary Note 1). The extrinsic noise is simulated as cell-to-cell variations in the protein translation rate $\nu_1$ with different coefficients of variation (CV). Parameters for the two-stage model are: $\nu_0 = 2.25$ h$^{-1}$, $d_0 = 1.125$ h$^{-1}$, $d_1 = 0.09$ h$^{-1}$ and $<\nu_1> = 45$ h$^{-1}$. In the case of no extrinsic noise (Var($\nu_1$) = 0 h$^{-2}$, blue triangles), the auto-correlation of the two-stage model approaches that of the birth-death model. With increasing extrinsic noise (Var($\nu_1$) = 100 h$^{-2}$, red crosses) the auto-correlation increases. The reason for this is that the covariance and the variance become dominated by the extrinsic noise, for which a much longer correlation time was assumed

systems including bacteria and human tissue cultures[17–19]. In a first experiment, we aimed to detect a temporal correlation of protein expression in intact plant leaves by determining the correlation of protein levels between different time points (auto-correlation)[20]. Towards this end we developed the following experimental setup: (1) We decided to compare the protein levels at only two time points because the experiments have to be done with excised leaves and prolonged maintenance is expected to produce artefacts. (2) Under steady state expression, we had difficulties to detect relative differences of protein levels within 3 h time intervals. We therefore used the photoconvertible NLS-KikGR. KikGR can be irreversibly converted from a green fluorescent protein (KikG) to red fluorescent protein (KikR) by 405 nm illumination[21]. Using this system we determined the production of new proteins[22] by converting KikG to KikR followed by the quantification of newly produced green fluorescent KikG after 3 and 6 h. (3) We targeted the fluorescent protein to the nucleus by adding a NLS sequence to facilitate the selection of single cells. (4) We expressed NLS-KikGR under the strong ubiquitously and constitutively active cauliflower 35S and the UBIQUITIN10 (UBQ10) promoters. Fairly strong constitutive promoters were choosen to reach sufficiently high expression levels and thereby fluorescence intensities to measure fluctuations. Although this limits general conclusions, this procedure should result in a conservative estimation of intrinsic noise in our experiments as experimental data and theoretical considerations show that constitutive promoters show the lowest intrinsic noise[8,23]. Two different promoters were selected to exclude that we are exploring a specific property of one promoter. (5) We excluded that movement of NLS-KikGR between cells leads to correlation between neighbouring cells by using a KikGR protein version that forms tetramers[21] which should not move between cells. We confirmed this by expressing the KikGR protein in single epidermal *Arabidopsis* cells by biolistic transformation. In these experiments we found no fluorescence in the neighbouring cells (Supplementary Fig. 1d-e[21,24]).

For the temporal correlation analysis, transgenic *p35S:NLS-KikGR* and *pUBQ10:NLS-KikGR Arabidopsis* leaves were dissected and kept in darkness for 36 h to reduce the amount of already converted red NLS-KikR protein. NLS-KikG was converted to the NLS-KikR by confocal laser scanning microscopy (CLSM). The amount of KikG was determined at three time points (0 h, 3 h and 6 h, Figs. 1a, b). To minimize technical errors and to control bleaching effects we measured each nucleus at each time point two times and used the mean for further calculations. We used at least three biological replicas to determine the average Spearman and Pearson's correlation coefficients of the fluorescence levels between the 3-h intervals: *p35S:NLS-KikGR* (number of leaves = 4, $n = 393$ cells, Spearman's: $r = 0.83$, Pearson's: $r = 0.88$, example leaf: Fig. 1c, Supplementary Fig. 2), *pUBQ10:NLS-KikGR* (number of leaves = 3, $n = 153$ cells, Spearman's: $r = 0.76$, Pearson's: $r = 0.80$, example leaf: Fig. 1d, Supplementary Fig. 3). Control experiments with *p35S:NLS-KikGR* plants without the 36 h dark treatment exhibited fluctuations in a similar range (number of leaves = 10, $n = 465$ cells, Spearman's: $r = 0.59$, Pearson's: $r = 0.68$, Supplementary Fig. 4). The finding that the correlations coefficients were always clearly below 1 (perfect correlation) indicates that we can detect fluctuations between the two 3-h time intervals.

In order to put the experimentally determined correlation coefficients into a context we used a modelling approach aiming to address two questions: Is the linear two-stage model shown in Fig. 2 a sufficient to explain the data? How does cell-to-cell variability affect the decay of the auto-correlation?

Towards this end, we analytically calculated the non-stationary auto-correlation function of the linear two-stage model with a

stochastic translation rate $v_1$ as a source of extrinsic noise (Supplementary Note 1). We further simulated the stochastic KikGR system. In Fig. 2b we show example trajectories before and after the converting light pulse. In order to make a prediction for the value of the temporal auto-correlation between 6 h and 3 h after the conversion we need to obtain estimates for the model parameters. We estimated the degradation rate $d_1$ of KikR using the measured values of the red fluorescent protein at 3 h and 6 h after conversion (Supplementary Note 1; $d_1 = 0.09\,\text{h}^{-1} \pm 0.023\,\text{h}^{-1}$). The other model parameters are unknown and it is not easy to obtain reliable estimates. However, we can show that the auto-correlation for the non-stationary two-stage process with extrinsic translational noise is bounded from below by the much simpler auto-correlation function of the one-stage death-birth process (taking only protein production and decay into account), which only depends on the stability of the protein (Fig. 2c, Supplementary Note 1, Supplementary Fig. 24). According to this we estimated the value $r$ for auto-correlation of the KiKG gene

expression between 3 h and 6 h to be in the range $1 \geq r \geq 0.6$ (Supplementary Note 1). Our experimental data are consistent with this expectation suggesting that the two-stage model provides a good estimate for the underlying noise. Cell-to-cell variability prolongs the auto-correlation time, given that the correlation time of the extrinsic noise is longer than the correlation time of the intrinsic fluctuations (an assumption underlying our analytical calculations, Supplementary Note 1).

**Extrinsic and intrinsic noise in different tissues**. To enable a spatial analysis of the intrinsic and extrinsic noise we adopted a dual reporter strategy initially used in bacteria and yeast (Fig. 3a) [8,12]. Extrinsic noise is seen when both marker values correlate and show the same variation. Intrinsic noise is recognized when the two marker values are not correlated in single-cell measurements. We generated transgenic plants expressing 2xNLS-YFP and 2xNLS-CFP under the control of the 35S promoter. We used YFP and CFP fusions to two nuclear localization signals (2xNLS) for two reasons. First, by targeting the signal to one defined region in the cell, the nucleus, we improved the accuracy of measurements. Second, the targeting of the marker to the nucleus reduces the intercellular mobility that would lead to an underestimation of fluctuations [24] (Supplementary Note 2, Supplementary Fig. 1a-c, e).

The analysis of pavement cells in young and mature rosette leaf stages revealed intrinsic and extrinsic noise. As shown in Fig. 3b the colour of individual nuclei ranged from green to magenta in merged YFP/CFP pictures indicating that the relative expression of the two 35S promoters driving 2xNLS-CFP and 2xNLS-YFP differs from cell to cell. This is indicative for intrinsic noise. Plotting the mean CFP values against the mean YFP values revealed intrinsic and extrinsic noise for young and mature leaves (representative leaf shown in Fig. 3c, d, Supplementary Figs. 5 and 6). The statistical analysis revealed significantly higher extrinsic noise than intrinsic noise (Supplementary Note 3, Fig. 3e, f, young leaf: $p = 1.1 \times 10^{-5}$ and mature leaf: $p = 7.6 \times 10^{-5}$, Wilcoxon rank-sum test). Thus, extrinsic noise is the major source of noisy gene expression in young and mature rosette leaves. This parallels previous findings in yeast [2,20,25].

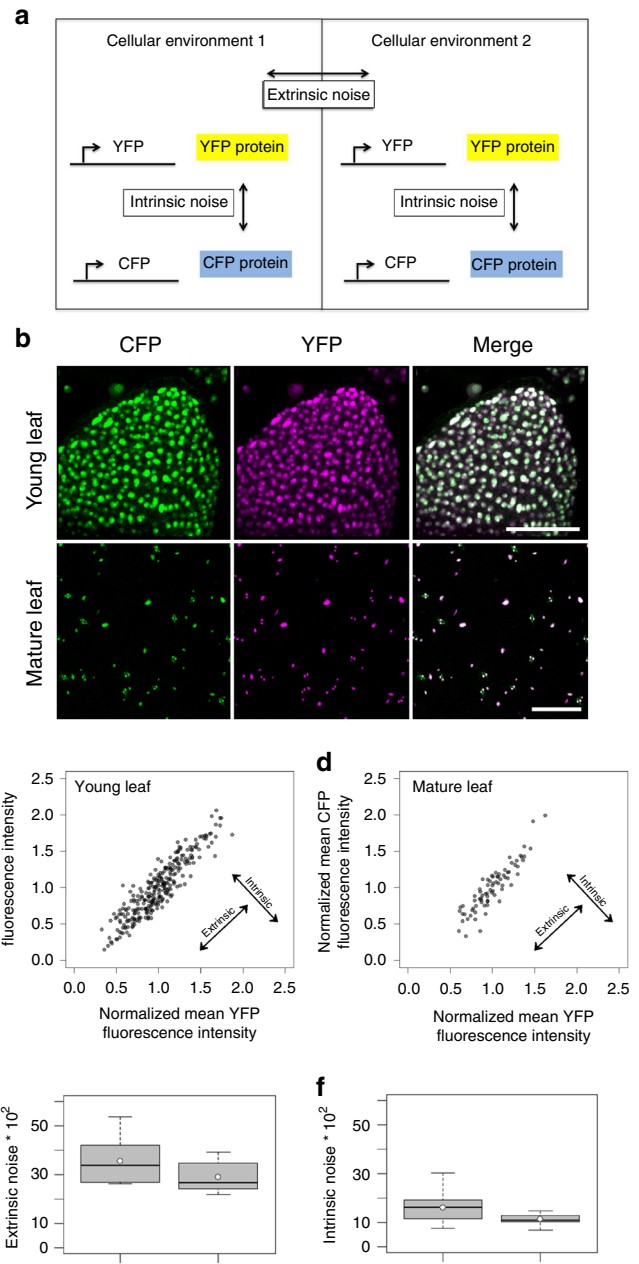

**Fig. 3** Intrinsic and extrinsic noise in young and mature rosette leaves of *p35S:2xNLS-YFP p35S:2xNLS-CFP* plants. **a** Schematic illustration of the experimental setup to determine the intrinsic and extrinsic noise. **b** CLSM images of a young, developing leaf and a mature leaf of a *p35S:2xNLS-YFP p35S:2xNLS-CFP* line. CFP is shown in green, YFP in magenta and the same fluorescence levels of both is indicated in white. Note, stomata show autofluorescence in the CFP channel. Scale bars: 50 μm (young leaf) and 100 μm (mature leaf). **c** Scatter plot of the normalized CFP mean fluorescence intensity plotted against the normalized YFP mean fluorescence intensity of single cells in one representative young leaf (*n* = 284). Pearson's correlation coefficient = 0.914, Spearman's correlations coefficient = 0.905. Data points are shown in grey, overlapping data points appear black. **d** Scatter plot of the normalized CFP mean fluorescence intensity plotted against the normalized YFP mean fluorescence intensity of single cells in one representative mature leaf (*n* = 76). Pearson's correlation coefficient = 0.909, Spearman's correlation coefficient = 0.906. **e** Box plot of extrinsic noise measurements of young (*n* = 10 leaves with a total number of 2219 cells, median = 33.5) and mature leaves (*n* = 10 leaves with a total number of 757 cells, median = 26.6). The extrinsic noise was slightly but not significantly higher in young leaves as compared to mature leaves (*p* = 0.075 Wilcoxon rank-sum test). **f** Box plot of intrinsic noise measurements of young (*n* = 10 leaves with a total number of 2219 cells, median = 16.1) and mature leaves (*n* = 10 leaves with a total number of 757 cells, median = 10.8). The intrinsic noise was significantly higher in young leaves (*p* = 0.029 Wilcoxon rank-sum test). Boxes show 25th and 75th percentiles and median. White dots show mean values

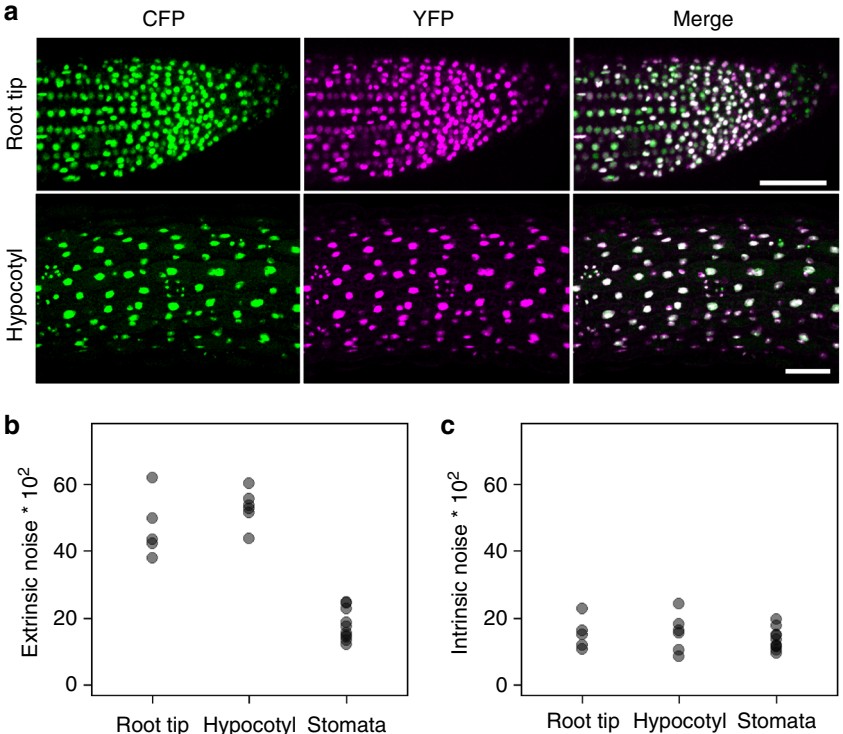

**Fig. 4** Intrinsic and extrinsic noise in stomata cells, root tip cells and hypocotyl cells of *p35S:2xNLS-YFP p35S:2xNLS-CFP* plants. **a** CLSM images of a root tip and a hypocotyl of *p35S:2xNLS-YFP p35S:2xNLS-CFP* plants. CFP is green, YFP is magenta and overlay is white. **b** Plot of extrinsic noise of root tip cells ($n = 6$ roots with a total number of 463 cells, median = 43.4), hypocotyl cells ($n = 6$ hypocotyls with a total number of 690 cells, median = 53.1) and stomata cells ($n = 10$ from mature leaves with a total number of 513 cells, median = 16.6). **c** Plot of intrinsic noise of root tip cells ($n = 6$ roots with a total number of 463 cells, median = 15.1), hypocotyl cells ($n = 6$ hypocotyls with a total number of 690 cells, median = 15.9) and stomata cells ($n = 10$ from mature leaves with a total number of 513 cells, median = 12.8). The extrinsic noise in root tip cells and hypocotyl cells was significantly higher than in stomata cells ($p = 0.00067$ and $p = 0.00025$, Wilcoxon rank-sum test)

These conclusions were confirmed using an independently transformed *Arabidopsis* line carrying the *p35S:2xNLS-YFP p35S:2xNLS-CFP* constructs (Supplementary Figure 7 a-b, Supplementary Figure 8, Supplementary Figure 9). To test whether the high extrinsic noise is specific to the 35S promoter, we also tested the *UBQ10* promoter. Stably transformed *pUBQ10:2xNLS-YFP pUBQ10:2xNLS-CFP Arabidopsis* plants revealed similar behaviour as described for the *35S* promoter (Supplementary Fig. 7c-e, Supplementary Fig. 10, Supplementary Fig. 11).

These findings raised the question whether noise differs in different cell types or tissues. We therefore determined the noise additionally in stomata, epidermal hypocotyls and root tip cells of *p35S:2xNLS-YFP p35S:2xNLS-CFP* plants (Fig. 4, Supplementary Figs. 12–18). Intrinsic and extrinsic noise were found in a similar range in all tissues/cell types except for stomata. For stomata we found a clearly and significantly lower extrinscic noise in both independent transgenic lines (root/ stomata: $p = 0.0007$, $p = 0.03$, hypocotyl/stomata: $p = 0.0002$, $p = 0.03$, pavement cells in young leaves/stomata ($p = 1.08 \times 10^{-5}$, $p = 0.006$, Wilcoxon rank-sum test). This indicates that extrinsic noise can vary in a tissue/cell type specific manner.

**Noise in cells with different DNA contents**. Next we tested the concept whether higher endoreduplication levels lead to reduced noise. We took advantage of the fact that pavement cells exhibit a wide range of ploidy levels between 2C and 64C[4,26]. This allowed us to study the correlation between ploidy and noise for one specific cell type. As higher DNA contents lead to increased nuclear sizes, we used the maximal area of each nucleus as an

estimator for the DNA content in our correlation studies[27]. We determined the maximal nuclear area of leaf epidermal cells in a stack of images and analyzed the YFP and CFP values. We estimated the median of nuclear area of all pavement nuclei and considered four quartiles separately in *p35S:2xNLS-YFP p35S:2xNLS-CFP* and *pUBQ10:2xNLS-YFP pUBQ10:2xNLS-CFP* plants. Intrinsic noise levels were similar in all four quartiles (Supplementary Fig. 19 b–e; Supplementary Fig. 20b). This finding is not unexpected as already two gene copies in a diploid cell might be sufficient to balance fluctuations in one of them. For extrinsic noise we observed the trend that larger nuclei have slightly more extrinsic noise (Supplementary Fig. 19c,f; Supplementary Fig. 20c). Increased extrinsic noise in larger cells could be explained by less uniform cellular states in cells with a higher DNA content or by changes due to a progression of cell differentiation.

**Correlation of noise between neighbouring cells**. Finally, we aimed to understand whether fluctuation in gene expression is coupled in neighbouring cells. In contrast to unicellular bacteria or yeast one could envision that in tissues extrinsic noise might be correlated in immediately neighbouring cells. This could result from initially similar cellular conditions in daughter cells or cellular connectivity of plant cells by plasmodesmata. To exclude that the low movement rates of the fluorescent marker protein used in this study leads to a correlation between neighbouring cells we estimated the transport coefficient for the mobility between cells. We found that the contribution to a cell-cell correlation due to mobility of the fluorescent protein is negligible

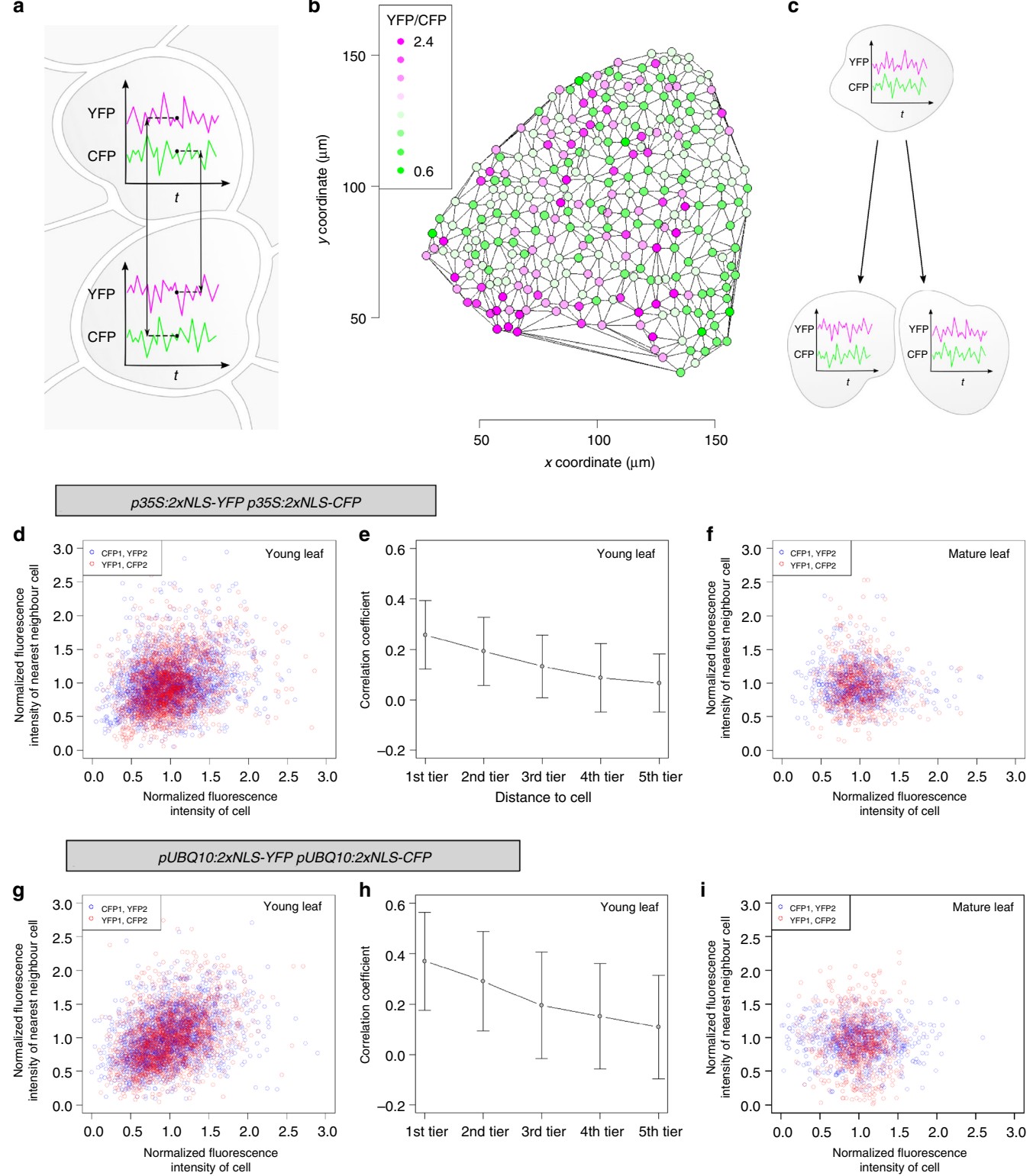

(Supplementary Note 2). We reasoned that the intrinsic gene expression noise is mechanistically decoupled between cells, i.e., the expression of the fluorescent protein in one cell does not influence the expression in another cell. Moreover, due to cell division the stochastic gene expression in the growing tissue is never in stationary state. This would yield an extra contribution from the intrinsic noise if one would analyze the spatial correlation using a single reporter system (Supplementary Note 2).

However, using the dual reporter *p35S:2xNLS-YFP p35S:2xNLS-CFP* plants for a cross-analysis (relating CFP to YFP and vice versa, see Fig. 5a) we can estimate the variance of the extrinsic noise (e.g., variability in ribosome number, transcription factor abundance[28,29]) and the covariance between the extrinsic noise of neighbouring cells (Fig. 5b). The covariance between stochastically identical cells is equal to the variance of the extrinsic noise. Therefore, it is necessary to normalize the covariance using the

variance of the extrinsic noise to obtain a measure between 1 and −1 for the correlation between neighbouring cells (Supplementary Note 2). In this way we estimated the correlation of the extrinsic noise between nearest neighbour cells in young leaves and found a weak but significant correlation ($r = 0.34$, $p < 0.0002$, randomization test, Fig. 5d). To test whether this correlation ceases with increased distances we calculated the correlation between each nucleus and its 39 closest neighbours. We observed a drastic reduction with increasing distance. To judge over how many cell diameters extrinsic fluctuations are correlated we determined the average nearest neighbour distance and used this value to define five concentric rings (tiers) of cell distances (Fig. 5e). These data indicate that on young leaves correlation is mainly found between immediately neighbouring cells. By contrast, we detected no correlation between neighbouring cells on mature leaves ($r = 0.02$, $p = 0.433$, randomization test, Fig. 5f). Similar results were obtained with the second p35S:2xNLS-YFP p35S:2xNLS-CFP line ($r = 0.423$; $p = 0.0014$; Supplementary Fig. 21) and with a pUBQ10:2xNLS-YFP pUBQ10:2xNLS-CFP line ($r = 0.413$; $p = 0.0003$, randomization test, Fig. 5g–i). Thus, correlation is only found in young but not in mature leaves (see confirmation in independent transformants in Supplementary Figure 21). A distance dependent correlation of extrinsic noise was also found in hypocotyl and root tissues for two independent p35S:2xNLS-YFP p35S:2xNLS-CFP lines (Supplementary Fig. 22, Supplementary Fig. 23).

To judge to what extent inheritance of mRNA and protein content can explain the observed next-neighbour correlation we used the two-stage gene expression model shown in Fig. 1 a under the following assumptions: At time $t = 0$ the mRNA and protein content of a mother cell expressing the dual reporter system is copied to two daughter cells. Thus daughter cells have identical initial condition for the mRNA and protein amount. Except for the translation rate the cells inherit all parameter values from the mother cell (Fig. 5c). The translational rates of the daughter cells are stochastic and in general different from each other (extrinsic noise). Given the observed division rate of Arabidopsis leaf epidermal cells and the degradation rate of the reporters we estimated the contribution of mRNA and protein inheritance to the next-neighbour correlation to be $r \approx 0.16$ (Supplementary Note 4, Supplementary Fig. 25). This indicates that the inheritance of mRNA and protein content is not sufficient to explain the observed spatial correlation. To estimate the maximal correlation caused by inheritance we considered the case that not only the mRNA and protein content is inherited but also all rates related to gene expression (i.e., all rates of the daughter cells are

identical and equal to the rates of the mother cell and do not change over time). In this case we found a correlation of $r = 0.4$ which is close to the experimentally observed value.

## Discussion

As reported before in bacteria, yeast and animals we report in this manuscript fluctuations of gene expression in 3 h time intervals. Interestingly, a recent publication by Meyer et al.[6], provided evidence that such a temporal fluctuation can be fixed and translated into different cell differentiation responses.

The use of a dual reporter system enabled us to distinguish between extrinsic and intrinsic noise. Consistent with results in yeast[2,20,25], we found that extrinsic noise is the major source of noisy gene expression in young and mature rosette leaves. This indicates that the physiological state of plant cells equally affecting expression of all genes creates more noise than the intrinsic stochasticity of molecular processes influencing transcription and translation. It is therefore conceivable that differences of extrinsic noise in different cell types reflects different physiological properties or states of the cell types. Consistent with this, our theoretical analysis of the spatial correlation suggests, that the inheritance of mRNA and proteins is not sufficient to explain the spatial correlation and that the inheritance of cellular conditions (e.g., ribosome number, stress status) and/or cell-cell communication is required. This explanation also fits the finding that we found no spatial correlation in mature leaves as both, cell division rates[30] and the number of open plasmodesmata ceases in mature tissues[31]. In this light, spatial coupling of extrinsic noise suggests that some processes needed for gene expression co-vary. One possible explanation is that tissues are composed of micro-domains with different physiological properties that lead, e.g., to different numbers of accessible RNAs and/or ribosomes.

## Methods

**Construct generation**. pENS-YFP GW (Glyphosat^R) and pENS-CFP GW (Gly-phosinat^R) were modified by introducing the phosphorylated linker SalI-SV40NLS-XhoI (5′-CTCGAGATGCCAAAGAAGAAAAGAAAAGTTGAA-GATCCTGGGTCGAC-3′) into the XhoI restriction site of the vectors. For generation of pENS-2xNLS-YFP GW and pENS-2xNLS-CFP GW the ligation procedure was repeated and an additional SV40NLS was introduced into the XhoI site. In order to introduce a stop codon downstream of the YFP sequence an LR reaction (Gateway® cloning of system Invitrogen) was performed with pENTR1A-ccdB[33] and the destination vectors. p35S:NLS-KikGR and pUBQ10:NLS-KikGR were generated by LR reactions using pENTRA-NLS-KikGR and the pAMPAT plasmids.

All constructs and their use in different experiments are summarized in Supplementary Table 1.

**Fig. 5** Nearest neighbour analysis of p35S:2xNLS-YFP p35S:2xNLS-CFP and pUBQ10:2xNLS-YFP pUBQ10:2xNLS-CFP plants. **a** Schematic illustration of the experimental setup to determine the cofluctuation of neighbouring cells. **b** Leaf area depicting the cell-to-cell variability of noise based on the YFP/CFP ratios in each cell. Colours show the YFP/CFP ratios as indicated in the legend. **c** Schematic illustration of the effect of cell division on cofluctuation. **d** Scatter plot of p35S:2xNLS-YFP p35S:2xNLS-CFP young leaves showing the normalized fluorescence intensities of cells plotted against the normalized fluorescence intensity of the nearest neighbour of the considered cells (neighbour cell with the lowest distance). Blue circles indicate the CFP fluorescence intensity of a cell (CFP1) plotted against the YFP fluorescence intensity of the nearest neighbouring cell (YFP2). Red circles show the YFP fluorescence intensity of a cell (YFP1) plotted against the CFP fluorescence intensity of the nearest neighbouring cell (CFP2) ($n = 2219$ cells; $r = 0.341$; $p = 0.0002$, randomization test). **e** Mutual dependency of the distance to the neighbouring cell and the cofluctuation in young rosette leaves of p35S:2xNLS-YFP p35S:2xNLS-CFP. Neighbouring cells were grouped into five tiers according to their distance (cell diameters) to the considered cell. Mean values and standard deviations are shown ($n = 86,541$ neighbourhood analyses (2219 cells×39 cells)). (**f**) Scatter plot of p35S:2xNLS-YFP p35S:2xNLS-CFP mature rosette leaves showing the normalized fluorescence intensities of cells plotted against the normalized fluorescence intensity of the nearest neighbour ($n = 757$ cells; $r = 0.02$; $p = 0.433$, randomization test). **g** Scatter plot of pUBQ10:2xNLS-YFP pUBQ10:2xNLS-CFP young rosette leaves showing the normalized fluorescence intensities of cells plotted against the normalized fluorescence intensity of the nearest neighbour ($n = 2021$ cells; $r = 0.413$; $p = 0.0003$, randomization test). **h** Mutual dependency of the distance to the neighbouring cell and the cofluctuation in young rosette leaves of pUBQ10:2xNLS-YFP pUBQ10:2xNLS-CFP. Neighbouring cells were grouped into five tiers according to their distance (cell diameters) to the considered cell. Mean values and standard deviations are shown ($n = 78,819$ neighbourhood analyses (2021 cells×39 neighbouring cells)). (**i**) Scatter plot of pUBQ10:2xNLS-YFP pUBQ10:2xNLS-CFP mature rosette leaves showing the normalized fluorescence intensities of cells plotted against the normalized fluorescence intensity of the nearest neighbour ($n = 775$ cells; $r = −0.06$; $p = 0.681$, randomization test)

**Transient transformation by particle bombardment**. *Arabidopsis* leaves were transiently transformed by particle bombardment using a particle gun (gene gun). 0.8 μg DNA of each construct were used and pipetted into one reaction tube. 10 μl gold (30 mg/ml; diameter 1 μm), 20 μl CaCl$_2$ 2.5 M and 8 μl spermidine 0.1 M were added. After incubation for 10 minutes at room temperature the gold suspension was centrifuged (10 s, 10,000 r.p.m.) and the coated gold particles were resuspended in 100 μl 70% EtOH. After a second centrifugation step (10 s, 10,000 r.p.m.) 50 μl absolute EtOH were added to the gold particle pellet. After resuspending the pellet, the suspension was centrifuged again (10 s, 10,000 r.p.m.). Finally, the gold particle pellet was resuspended in 15 μl abs. EtOH and placed onto a plastic disc (macro carrier). After drying, the macro carrier was placed into the particle gun and the gold particles were used for bombardment of leek cells. Rupture disks (900 psi) and a vacuum of 26 Hg (inch of mercury, equivalent to 3.38 Pa at 0 °C) were applied during bombardments. After bombardment the samples were stored in darkness at room temperature and were analyzed 16–24 h after the transformation procedure.

**Stable transformation of *Arabidopsis thaliana***. 5 ml pre-culture of agrobacteria containing the desired construct were grown over night. At the next day 200 ml were inoculated with 500 μl of the pre-culture. After ~24 h 10 g sucrose and 50 μl Silwet L-77 were added to the culture. Plant flowers were dipped into the suspension for 10 s. For double transformation 100 ml of each culture were mixed shortly before transformation as described previously[32]. 10 g and 50 μl Silwet L-77 were added and flowers were incubated in the suspension for 10 s.

**Confocal laser scanning microscopy**. Confocal laser scanning microscopy (CLSM) images were generated using Leica TCS SPE. Images were analyzed and quantified using the software ImageJ. Mean grey values (0 = black; 255 = white) of regions of interest (ROIs) of 8 bit images were used for calculations. Analyses were always performed with overlaying maximum Z-stack projection images. Laser, gain, and detection parameters were never changed for all image acquisitions (Supplementary Table 2).

**Measurement of photoconvertible NLS-KikGR**. Single leaves (leaf number 3 or 4) of 7 days old stably transformed *p35S:NLS-KikGR* and *pUBQ10:NLS-KikGR* plants (Col-0) were imaged by CLSM (Leica TCS SPE). The leaves were placed onto 1% MS agar on a cover slip for imaging. NLS-KikG was photoconverted to NLS-KikR by a 405 nm laser line (100% laser power) exciting the sample for 5 to 10 s. NLS-KikR degradation and re-synthesized NLS-KikG were sequentially imaged two times at three different time points (0, 3, and 6 h) with the defined Z-slide distance of 3.0 μm. Each nucleus was manually selected from Z-stack projections (512px×512px) of entire leaves to find the maximal cross-section. The boundary of the nucleus was always clearly seen independent of the fluorescence intensity. Subsequently, we determined the mean grey intensity of the selected region using ImageJ. Mean grey values of ROIs were used for calculation of protein degradation of NLS-KikR and protein synthesis of NLS-KikG.

To test whether NLS-KikGR can move between cells, we transformed single *Arabidopsis* leaf epidermal cells by biolistic transformation with *p35S:NLS-KikGR*. Among 20 transformed cells we found not a single case where fluorescent signal could be detected in the neighbouring cells (Supplementary Fig. 1d-e).

**Measurement of CFP and YFP fluorescence intensities**. Stably transformed *p35S:2xNLS-YFP p35S:2xNLS-CFP* and *pUBQ10:2xNLS-YFP pUBQ10:2xNLS-CFP* plants (Col-0) were imaged by CLSM (Leica TCS SPE). Laser, gain and detection parameters were never changed for all image acquisitions. CFP and YFP fluorescence intensities were sequentially imaged with the defined Z-slide distance of 1.51 μm. Nuclei were selected and analyzed as describe above. The mean background of each channel and image was measured separately and subtracted from the CFP and YFP mean grey values and the data were normalized using the mean fluorescence of the data set. For raw images see Supplementary Fig. 26. Calculations of extrinsic and intrinsic noise were performed for each image separately and finally all noise values of each image of the same tissue were presented in a box plot including the corresponding median and mean values. We excluded samples from the analysis in which the YFP and CFP value distributions were significantly different in a Kolmogorov–Smirnov test to exclude that a skewing between the two channels influences our analysis. For all statistical analysis we confirmed that the data structure is adequate. Each root tip was virtually rotated in the $z$–$y$ axis to ensure a horizontal position of the root tip in the image. Only the upper 15 μm layer was analyzed for calculations of noise to select only epidermal cells.

**Code availability**. The codes that support the findings of this study are available from https://gitlab.com/wurssb/Stochastic_GE_in_Arabidopsis_thaliana.git.

**Data availability**. Additional data that support the findings of this study are available on request.

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

## Acknowledgements

We like to thank Dr Alexandra Steffens for providing the UBQ10 promoter sequence, Irene Klinkhammer for technical help, Dr A. Schauss, CECAD Imaging facility, for helping us to provide all relevant controlls on the microscopic noise. This project was supported by the Deutsche Forschungsgemeinschaf priority programme SFB 572 and SFB 680. R.B. was funded by the International Graduate School in Genetic and Functional Genomics (IGS-GFG), University of Cologne.

## Author contributions

I.S.A, C.F. and M.H. designed the project; I.S.A, J.M.P. and R.B. performed the biological experiments, E.M.K., B.G. and C.F. did the theoretical studies, and I.S.A., M.H. and C.F. wrote the manuscript.

## Additional information

**Competing interests:** The authors declare no competing financial interests.

