## [Peer Review File · Nature Communications]

Reviewers' comments:

Reviewer #1 (Remarks to the Author):

The manuscript from Schultheiß et al explores in detail transcriptional stochasticity in *Arabidopsis thaliana*. The authors first use a photoconvertible marker to analyse fluctuation of gene expression and then a dual reporter system to compare intrinsic and extrinsic noise of individual cells. They show that extrinsic noise is the major source of transcriptional stochasticity in all tissues tested. Their results also indicate that stochastic gene expression is coupled for cells in young leaves but not in mature leaves. The results are interesting, although it would be useful if the authors could make better links between the different experimental sections, and further attempt to make explicit the biological relevance of their results.

Major Comments:

The authors use a biolistic approach to transiently transform *Arabidopsis* leaves with p35S:NLS-KikGR or pUBQ10:NLS-KikGR. It would be good if the authors could explain how they controlled the amount of DNA introduced into each cell and how they ensured that this method is not creating noise in the amount of copies of p35S:NLS-KikGR or pUBQ10:NLS-KikGR between cells.

P2 57-59: "For fluctuation measurements transgenic p35S:NLS-KikGR *Arabidopsis* leaves were dissected and kept in darkness for 36 hours to reduce the amount of already converted red NLS-KikR protein". It would be good if the authors could include a control to define the effect on expression of the fact the leaves were dissected for 36 hours as this could be a potential stress for the leaf. Is it possible to show that dissection and darkness are not modulating noise in gene expression?

To solve the issues rose in points 1. and 2. and support their results, perhaps the authors could also analyse fluctuation of gene expression in stably transformed plants with p35S:NLS-KikGR or pUBQ10:NLS-KikGR, or discuss why this is not necessary?

It seems that there is a skewing between CFP and YFP signal at high intensities in the mature leaves in Figures 2C and S2A. It would help if the authors could provide the distribution of CFP and YFP fluorescence intensities distribution to ensure that they are similar in a given tissue. If the distributions of CFP and YFP are different (as it appears) how is this taken into account in the estimation of intrinsic or extrinsic noise?

In Figure 3 and S4, the authors concluded that extrinsic noise is higher in root tips and hypocotyls than in leaves and stomata. However, they don't comment on the very interesting profile they have in the root with 2 to 3 sub-populations that can be clearly identified. In Figure 3A, a first population can be observed at the level of the columella (purple in the merge image), another one at the level of the meristematic zone (white in the merge image) and a third one higher in the root tip (green in the merge image). Also, it would be better if the authors could change their conclusion p4 113-114: "These data indicate that extrinsic noise can vary in a cell type specific manner." as what they observe is that extrinsic noise can vary in a tissue specific manner, not a cell type specific manner, as there are different cell types in the root tip they are analyzing.

P4 126-128: "We found a subtle but significant correlation between nuclear size and extrinsic noise in mature leaves of p35S:2xNLS-YFP p35S:2xNLS-CFP and pUBQ10:2xNLS-YFP pUBQ10:2xNLS-CFP". The authors conclude this by comparing the 1st and 5th quintile, corresponding to the smallest and largest cells for p35S:2xNLS-YFP p35S:2xNLS-CFP and pUBQ10:2xNLS-YFP pUBQ10:2xNLS-CFP. However they do not comment about the fact that the 5th quintile is showing a very different behavior for the 2 promoters (Figures S5B and E).

Minor comments:

Figures 2D and 2E are inverted in the legend.

P4 118-119: "As higher DNA contents lead to increased nuclear sizes, we used the nuclear size as an estimator for the DNA content in our correlation studies". Even if this is commonly known, it would be good if the authors could cite a paper supporting it.

It would be nice if the authors could expend a little bit more on the fact they observe that neighbouring cells are more synchronised in young leaves than mature leaves. This is one of the most interesting results of the paper and could have been more discussed. To go further, could the authors perform the same analysis on other cell types that are known to have high or low levels of connection?

Reviewer #2 (Remarks to the Author):

As heterogeneity in gene expression has been shown to underlie stochastic cell differentiation in some species, it is quite possible that it is also responsible for the apparent developmental stochasticity in plants. In this paper Schultheiß and colleagues take a first step in the elucidation of this interesting question by investigating gene expression noise characteristics in *Arabidopsis thaliana*. As in all previously tested species, they find that gene expression is noisy, and that its extrinsic component is larger than its intrinsic. They quantify the extrinsic and intrinsic noise components for different cell types and in some cases find differences in the amount of extrinsic noise. Finally they investigate whether there is spatial coupling in the total gene expression of neighbouring cells, which they find for young leaf cells but not for old.

The manuscript presents an impressive amount of technically challenging experiments. In my opinion the claims are novel and will be of interest to others in the field. The unexpected claims (coupling in young leaf cells) could be more emphasised. And I am not convinced that all claims are supported by strong evidence.

** Major comments:

I believe there are some unsubstantiated claims in the manuscript.

- 20: I cannot find any data that shows significant differences in intrinsic noise between tissues
- 63: "cells producing high KikG levels in the first interval showed low levels in the second interval or vice versa (figure 1C)." This statement suggests a negative correlation, 2 sentences later it is claimed that there is no correlation, and in reality I believe there is a positive correlation.
- 67+78+156: It is very difficult to show that no correlation between two variables exists. The phrasing here ('The statistical analysis revealed no correlation', 'no correlation was found') can suggest that your data indeed showed that no correlation between these two variables exists. I actually expect that a student t-test will show that there is a significant correlation in the data of Fig1C, and perhaps even in Fig1D. I could not find a significance test for the claim that there is 'no correlation'. A student t-test or a Monte-Carlo significance test might help.

The main claims rely heavily on quantitative image analysis. However some methodological details are unclear, and some quantification should possibly be improved.

- Fig1C: KikG(6h)-KikG(3h) by itself does not only quantify the produced KikG in this time-period, but also the amount of KikG(3h)-KikG(0h) that was degraded in this time period. Did the authors corrected for this?
- 112: was autofluorescence subtracted from images? As mentioned in caption of Fig2 "stomata show

autofluorescence in the CFP channel". Was background fluorescence subtracted?

- 211: were all images taken below saturation level (255)? Some of the images suggest this may not be the case.
- 220+225: I am confused: both protein degradation of KikG and KikR are mentioned. What is measured and for what quantification is each measurement used? In the Supp Info protein half-life is measured by KikG degradation, but I would expect KikR would be used for that.
- 221: unclear whether Z-stack size depicts how far apart Z-slices are taken, or how far the total range is in which Z-slices are taken.
- 223 & FigS5: Unclear how nuclear sizes are determined. Note that in order to correlate nuclear sizes with their fluorescence (or noise) it is important that they are quantified independently (i.e. an over or underestimation of the nuclear size should not result in a systematic error in the fluorescence value).
- in order to quantify the intrinsic and extrinsic noise it is necessary to normalise the fluorescence data. However, due to normalisation of the data in the manuscript it is not possible to see differences in mean fluorescence levels between comparisons (for example between young and old leaflets in Fig2,4, FigS2-3, between different cell types in Fig3, FigS4, and between different nucleus sizes in FigS5). Can differences in extrinsic noise be explained by differences in mean fluorescence?

Some observations or data are not discussed.

- Fig2C, FigS2A, FigS3B: can the authors explain why in mature leaflets the relation between CFP and YFP does not lie on a $y=x$ line? Does this mean that the following does not hold: "The two fluorescent proteins exhibited statistically equivalent intensity distributions and thus displayed the necessary independence and equivalence to detect noise (7)." Elowitz 2002
- 110: I do not understand why this data is not (or cannot be) compared to the previous leaflet data.
- Fig3A: It seems that total expression (sum of CFP and YFP) in the root tips is highly dependent on the Z location. Could this be a systematic error in the measurement that would result in inflated extrinsic noise values?
- Fig4: These comparisons do no longer look at extrinsic and intrinsic noise. Why is the sum of YFP and CFP analysed? Does analysis of each fluorophore independently give the same result?
- Fig4: I do not understand how the correlation coefficient between the most similar cells (1st tier in C) is lower of that of the complete data set (B) which also contains the data of the most dissimilar cells.

** Minor comments:

- Abstract: I don't understand the logic of the introduction: the first sentence emphasises that development is reproducible despite molecular stochasticity, whereas the third sentence uses the fact that development is non-reproducible (stochastic) as a motivator for the study of molecular stochasticity.
- 16: 'maybe' > 'may be'.
- 17: 'reason' > 'source' / 'cause'?
- 19: 'expressions' -> not clear what is meant by this, gene expression / protein expression / phenotype expression?
- 33: 'sources' -> there are probably many sources, but the total noise can be 'divided in 2 components' (see Elowitz 2002).
- 34: 'of all'.
- 51: "n=20" I could not find the data, nor a positive control.
- 68 & 70: I get the impression that the correlation coefficient of the KikG production rates (68) is compared here to that of the total KikG amount (70). I would assume that the autocorrelation time of total KikG amount is set by the degradation / dilution rate, and rather independent of the KikG production rate.
- 69 + 72 + 92 + etc: please indicate where in Supp Info.
- 171-226: is repeated in Supp Info.

- 251: with $H_{int0} = 0.05$ and some H_{int} measurements near 0.05, I believe that some of the intrinsic noise values may be overestimated by much more than 14%.
- Fig1AB: Why are the resolution and quality of these images poorer than that of FigS1ABC?
- Fig1 contains data from multiple leafs. Is the normalisation done on each leaf separately, or on the combined data set?
- Fig3A: without the stomata cells data no large difference in extrinsic noise would be observed. Therefore it would be more convincing to show images of these cells too.
- Fig4: could the coupling be due to the recent common ancestry of the young leaf cells, instead of due to communication?
- SuppInfo 1.2: can the authors provide some intuition why minimizing function (1) estimates the degradation rate.
- FigS1-7: please let each figure be followed by its own legend.
- FigS3 legend: '100 m' -> '100 um', 'median=17.1' -> 'median=0.171'
- FigS5C-F: does this plot show the bootstrap analysis? It looks more like a box plot of the noise in each quintiles. The distributions of the 1st and 5th quintiles in C and E look very similar, are the p values correct?
- FigS6-7 legends: do not seem to correspond to their figures.
- 60: I don't believe that this technique where new protein production is quantified using a photoconvertible fluophore is well known. A reference to the paper that initialized the technique might be in place (Leung et al. Nat. Neurosci. 9, 1247-1256 (2006) or Raab-Graham et al. Science 314, 144-148 (2006)?)

Reviewer #3 (Remarks to the Author):

The ms "Stochastic Gene Expression in Arabidopsis" by Schultheiß et al. is an interesting paper on the important topic of gene regulation that is not addressed thoroughly enough in plants. The examination of the types of stochastic gene expression (intrinsic vs. extrinsic) is an important topic in asking how cells cope or take advantage of such stochasticity. The paper used some appropriate tools and analysis to ask that question. The quantitative techniques seem appropriate. My primary concern is the choice of promoters to study or perhaps it is a matter of better explaining the rationale for the choice. In my view, I think it would be enough to demonstrate the sources of noise in plant tissue and the underlying causes or consequences can be explored in future work. However, I thought the paper lacked adequate discussion in framing these results.

1. The authors chose to examine the 35S cauliflower mosaic virus promoter. This is very strong and non-endogenous promoter. The rationale was as follows:

"We chose this promoter as it is not obviously under the control of any plant specific pathway and should therefore enable us to study noise fairly independent from plant regulatory processes."

2. I can't quite understand the rationale. Why would you want to study noise independently of plant regulatory processes? I guess this means independently of trans regulatory processes but that is typically the source of extrinsic noise. Also, that doesn't quite make sense either because presumably the 35S promoter is hijacking the plant's own trans regulatory mechanisms to initiate transcription? I don't understand the rationale.

In fairness, the authors do a subset of experiments using the ubiquitin promoter. This is an endogenous promoter that is also expressed at high levels. I didn't see the rationale for choosing another regulatory region that mediates high expression.

3. There may be a good reason for choosing highly expressed regulatory regions but it would also seem to warrant some discussion of the limitations of this approach. Could intrinsic noise be more prominent for more lowly and moderately expressed genes such that the author's conclusions are limited to promoters mediating high rates of transcriptions. One could image that promoter activity is dependent on the concentration of trans factors and that would have an impact on noise? All this to say that it would be good to have more context for the rationale and results. In particular, the animal and yeast literature on this topic is quite extensive and it would have helped if the authors had drawn parallels for their choice of promoters, for example, to help us understand the rationale of the experiment and context for any conclusions. For example, I may be unaware of an aspect of that literature which really justifies the choice of reagents in this research and helps frame the result.

Overall, I thought the authors explored a great question with an interesting approach but they needed more rationale of their experimental set up and a discussion of its implications.

Reviewer #1 (Remarks to the Author):

1) The results are interesting, although it would be useful if the authors could make better links between the different experimental sections, and further attempt to make explicit the biological relevance of their results.

In the current manuscript we have now extended the descriptions and discussions at various places to put the results into a biological context. In particular we add a discussion including a theoretical analysis of the finding that noise is coupled between neighbouring cells.

Major Comments:

2) The authors use a biolistic approach to transiently transform Arabidopsis leaves with p35S:NLS-KikGR or pUBQ10:NLS-KikGR. It would be good if the authors could explain how they controlled the amount of DNA introduced into each cell and how they ensured that this method is not creating noise in the amount of copies of p35S:NLS-KikGR or pUBQ10:NLS-KikGR between cells.

We used transgenic plants carrying the p35S:NLS-KikGR or pUBQ10:NLS-KikGR constructs. This is now pointed out more explicit in the current manuscript. Therefore, the amount of DNA introduced into the cells is the same and cannot introduce noise.

We used transient expressions only to exclude that the KikGR protein can move between cells, which would lead to an underestimation of the correlation. To test whether KikGR can move from one cell into the neighbouring cell it is important to express the protein in only one cell. This enables to judge the mobility by analysing whether the neighbouring cells show also fluorescence. In these control experiments noise was not studied.

3) P2 57-59: "For fluctuation measurements transgenic p35S:NLS-KikGR Arabidopsis leaves were dissected and kept in darkness for 36 hours to reduce the amount of already converted red NLS-KikR protein". It would be good if the authors could include a control to define the effect on expression of the fact the leaves were dissected for 36 hours as this could be a potential stress for the leaf. Is it possible to show that dissection and darkness are not modulating noise in gene expression?

We repeated these experiments without the 36 hours dark treatment using the p35S:NLS-KikGR lines. The correlation between the two intervals was again very low in a similar range as observed before. These data are added to the manuscript.

4) To solve the issues rose in points 1. and 2. and support their results, perhaps the authors could also analyse fluctuation of gene expression in stably transformed plants with p35S:NLS-KikGR or pUBQ10:NLS-KikGR, or discuss why this is not necessary?

All experiments were done in stably transformed plants.

5) It seems that there is a skewing between CFP and YFP signal at high intensities in the mature leaves in Figures 2C and S2A. It would help if the authors could provide the distribution of CFP and YFP fluorescence intensities distribution to ensure that they are similar in a given tissue. If the distributions of CFP and YFP are different (as it appears) how is this taken into account in the estimation of intrinsic or extrinsic noise?

We appreciate this comment and followed the suggestion of the referee to analyse the CFP and YFP distribution in each tissue. We found that only very few samples displayed a skewing between CFP and YFP for unknown reasons. We therefore did not consider these samples in our analysis to ensure that our estimation of intrinsic and extrinsic noise is not biased because of a potential skewing effect.

6) In Figure 3 and S4, the authors concluded that extrinsic noise is higher in root tips and hypocotyls than in leaves and stomata. However, they don't comment on the very interesting profile they have in the root with 2 to 3 sub-populations that can be clearly identified. In Figure 3A, a first population can be observed at the level of the columella (purple in the merge image), another one at the level of the meristematic zone (white in the merge image) and a third one higher in the root tip (green in the merge image).

We initially also had this impression and tested the hypothesis by dividing the root into three or four regions in different manners. Unfortunately, when considering all roots we found no significant differences between the selected regions.

7) Also, it would be better if the authors could change their conclusion p4 113-114: "These data indicate that extrinsic noise can vary in a cell type specific manner." as what they observe is that extrinsic noise can vary in a tissue specific manner, not a cell type specific manner, as there are different cell types in the root tip they are analyzing.

We face the situation that our examples include specific cell types (stomata and pavement cells in young and mature leaves) and tissues in which different epidermal cell types are included (hypocotyl and root). We now use "tissues" or "tissues/cell types" depending on the context.

8) P4 126-128: "We found a subtle but significant correlation between nuclear size and extrinsic noise in mature leaves of p35S:2xNLS-YFP p35S:2xNLS-CFP and pUBQ10:2xNLS-YFP pUBQ10:2xNLS-CFP". The authors conclude this by comparing the 1st and 5th quintile, corresponding to the smallest and largest cells for p35S:2xNLS-YFP p35S:2xNLS-CFP and pUBQ10:2xNLS-YFP pUBQ10:2xNLS-CFP. However they do not comment about the fact that the 5th quintile is showing a very different behavior for the 2 promoters (Figures S5B and E).

The extrinsic noise was always higher in the largest nuclei populations than in the smallest in both 35S and the UBQ10 promoter lines. We do not comment on the finding that we find a different curve shape between the two promoters such that the 35S noise values continuously increase from the small to the large nuclei whereas the UBQ10 noise values appear to increase in the 2nd and 3rd quartile and drop in the 4th Quartile. We feel that the method chosen to judge the nuclei sizes is too crude to discuss differences between each quartile in a meaningful manner. We therefore comment only on the differences between the largest and smallest classes.

Minor comments:

9) Figures 2D and 2E are inverted in the legend.

The figures have been largely remade and in this context legends were also rewritten and adapted.

10) P4 118-119: "As higher DNA contents lead to increased nuclear sizes, we used the nuclear size as an estimator for the DNA content in our correlation studies". Even if this is commonly known, it would be good if the authors could cite a paper supporting it.

We added a reference reporting this correlation.

11) It would be nice if the authors could expend a little bit more on the fact they observe that neighbouring cells are more synchronised in young leaves than mature leaves. This is one of the most interesting results of the paper and could have been more discussed. To go further, could the authors perform the same analysis on other cell types that are known to have high or low levels of connection?

We expanded the analysis to hypocotyl and roots and also found a correlation of noise between immediately neighbouring cells. In addition, we add now a detailed analysis to judge the contribution of different aspects. In short, we introduce a model that considers the half-life time of GFP, the cell division rates and the two-stage gene expression model, which we also use to analyse the Kikume data, to estimate the contribution of cell division to the measured correlation. We find that inheritance of the mRNA and protein molecules would not be sufficient but that the inheritance of cellular states (for example the number of ribosomes etc) that influence noise could be sufficient. Another explanation of the observed correlation would be cellular communication via intercellular connections. The current data does not allow us to discriminate between the two possibilities or their relative contribution.

Reviewer #2 (Remarks to the Author):

** Major comments:

12) I believe there are some unsubstantiated claims in the manuscript.

- 20: I cannot find any data that shows significant differences in intrinsic noise between tissues

Thanks to the referee comments we took into account the skewing between YFP and CFP. This changed our results in a few cases. In particular we find now a statistically significant difference of intrinsic noise between pavement cells in young and old leaves.

13) -63: "cells producing high KikG levels in the first interval showed low levels in the second interval or vice versa (figure 1C)." This statement suggests a negative correlation, 2 sentences later it is claimed that there is no correlation, and in reality I believe there is a positive correlation.

We agree that our formulation suggested unintentionally a negative correlation and revised this part of the manuscript. During our revision, we realized a misunderstanding between the biologist and theoretical side. Instead of correlating the experimental measurements at 6 h and 3 h we correlated 6 h – 3 h and 3 h - 0 h. Using the correct analysis we find now an auto-correlation, which is in agreement with the theoretical predictions. We extended also our theoretical analysis from the birth-death model for the proteins to the two-stage model including mRNA and proteins, as well as extrinsic noise in the translational and transcriptional rates. From the theoretical considerations we find a minimum auto-correlation of $r=0.6$. Our experimental data is in agreement with these theoretical predictions. Details of the analysis are provided in the Supplement.

14) - 67+78+156: It is very difficult to show that no correlation between two variables exists. The phrasing here ('The statistical analysis revealed no correlation', 'no correlation was found') can suggest that your data indeed showed that no correlation between these two variables exists. I actually expect that a student t-test will show that there is a significant correlation in the data of Fig1C, and perhaps even in Fig1D. I could not find a significance test for the claim that there is 'no correlation'. A student t-test or a Monte-Carlo significance test might help.

Thank you for this comment. We revised this paragraph completely as described in the response above and provide both, the Spearman and the Pearson correlation coefficients.

15) -Fig1C: KikG(6h)-KikG(3h) by itself does not only quantify the produced KikG in this time-period, but also the amount of KikG(3h)-KikG(0h) that was degraded in this time period. Did the authors corrected for this?

In this figure (now figure 1D (35S promoter) and E (UBIQUITIN10 promoter)) we show the normalized difference of the mean fluorescence KikG intensity. In this graph we did not correct for our estimated degradation rate of KikG. However, we did consider this aspect in our models. Therefore we compare the experimental correlation coefficients with predicted values that do consider the degradation.

16) -112: was autofluorescence subtracted from images? As mentioned in caption of Fig2 "stomata show autofluorescence in the CFP channel". Was background fluorescence subtracted?

As the autofluorescence mentioned here was exclusively seen in the cell walls and as these did not overlap with the nuclei we did not subtract these values. Cytoplasmic background, however, was subtracted as it was done for all measurements.

17) -211: were all images taken below saturation level (255)? Some of the images suggest this may not be the case.

We strictly took images using settings that do not lead to saturation levels in the images. For the quantitative analysis these original images were used. The images used in the figures were corrected using the white balance option to make it easier for the general audience to recognize the structures.

18) -220+225: I am confused: both protein degradation of KikG and KikR are mentioned. What is measured and for what quantification is each measurement used? In the Supp Info protein half-life is measured by KikG degradation, but I would expect KikR would be used for that.

The referee is correct with the notion that we determined the degradation of KikR as this form cannot be synthesized by the cell. As KikR and KikG are still the same proteins, though with different fluorescence properties, we assumed that they do not differ in their degradation rate. We modified the corresponding parts.

19) -221: unclear whether Z-stack size depicts how far apart Z-slices are taken, or how far the total range is in which Z-slices are taken.

We changed this sentence to "with Z-Stacks taken at distances of 3 μm ". The range was always chosen so that nuclei were completely included in the stack.

20) -223 & FigS5: Unclear how nuclear sizes are determined. Note that in order to correlate nuclear sizes with their fluorescence (or noise) it is important that they are quantified independently (i.e. an over or underestimation of the nuclear size should not result in a systematic error in the fluorescence value).

Each nucleus was manually selected from a stack of images to find the maximal cross section. The boundary of the nucleus was always clearly seen independent of the fluorescence intensity. Subsequently, we determined the mean intensity of selected region. We changed the method section accordingly.

21) in order to quantify the intrinsic and extrinsic noise it is necessary to normalise the fluorescence data. However, due to normalisation of the data in the manuscript it is not possible to see differences in mean fluorescence levels between comparisons (for example between young and old leaves in Fig2,4, FigS2-3, between different cell types in Fig3, FigS4, and between different nucleus sizes in FigS5). Can differences in extrinsic noise be explained by differences in mean fluorescence?

Any scaling factor of the data is canceled when one calculated the extrinsic noise. In fact any linear transformation of the data does not affect the extrinsic noise. We explain this now in the Supplement.

Some observations or data are not discussed.

22) - Fig2C, FigS2A, FigS3B: can the authors explain why in mature leaves the relation between CFP and YFP does not lie on a $y=x$ line? Does this mean that the following does not hold: "The two fluorescent proteins exhibited statistically equivalent intensity distributions and thus displayed the necessary independence and equivalence to detect noise (7)." Elowitz 2002

see above point (5).

23) - 110: I do not understand why this data is not (or cannot be) compared to the previous leaf data.

We now add the leaf data to those of the other tissues and compare and include them in the text description in this paragraph.

24) Fig3A: It seems that total expression (sum of CFP and YFP) in the root tips is highly dependent on the Z location. Could this be a systematic error in the measurement that would result in inflated extrinsic noise values?

Thank you for this valuable comment. We re-analysed the data to exclude the possibility that the Z-location inflates the extrinsic noise values. In a first step we rotated each root in the Z-Y axis to ensure that the surfaces are exactly parallel. Next, we analysed only the upper 15 μm layer that safely includes all nuclei in the outer layer. For the noise analysis of the root we now use only these data.

25) Fig4: These comparisons do no longer look at extrinsic and intrinsic noise. Why is the sum of YFP and CFP analysed? Does analysis of each fluorophore independently give the same result?

We initially showed the scatterplot to visually describe the variation of the YFP and CFP levels by showing the sum. We agree with the referee that this is confusing as we did in fact all calculations with the cross-correlation between CFP and YFP. We now show the cross-correlation in the respective diagrams and add an extra diagram in figure 4A to illustrate the procedure.

26) Fig4: I do not understand how the correlation coefficient between the most similar cells (1st tier in C) is lower of that of the complete data set (B) which also contains the data of the most dissimilar cells.

The correlation that was shown in the scatterplot was mislabeled saying “normalized mean YFP+CFP fluorescence intensity of neighbor cell” instead of “normalized mean YFP+CFP fluorescence intensity of nearest neighbor cell” and correctly described in the figure legend. Sorry, for causing the confusion.

** Minor comments:

27) - Abstract: I don't understand the logic of the introduction: the first sentence emphasises that development is reproducible despite molecular stochasticity, whereas the third sentence uses the fact that development is non-reproducible (stochastic) as a motivator for the study of molecular stochasticity.

The abstract and the first part of the introduction have been rewritten.

16: 'maybe > 'may be'.

this has been corrected

- 17: 'reason' > 'source' / 'cause'?

this has been changed

-19: 'expressions' -> not clear what is meant by this, gene expression / protein expression / phenotype expression?

-
this has been changed to “protein expression”

-33: 'sources' -> there are probably many sources, but the total noise can be 'divided in 2 components' (see Elowitz 2002).

this has been changed as suggested

-34: 'of all'.

this has been corrected

28) 51: "n=20" I could not find the data, nor a positive control.

We have now added the details in the supplement. In Figure S1 we describe the NLS-KikGR mobility data together with the other YFP fusions proteins. The latter serve as a positive control in this context as they do show little movement. We analysed 20 transformed single cells and found not a single case in which we could detect a fluorescent signal in the neighboring cells.

29) -68 & 70: I get the impression that the correlation coefficient of the KikG production rates (68) is compared here to that of the total KikG amount (70). I would assume that the autocorrelation time of total KikG amount is set by the degradation / dilution rate, and rather independent of the KikG production rate.

The referee is correct in that we correlate the protein amounts (better fluorescence intensities) and not the production rates. We were not correctly describing this and rephrased this passage.

30) -69 (in 1.2) + 72 + 92 + etc: please indicate where in Supp Info.:

We indicate now in the text in which sections the Supplementary Information is reported.

31) -171-226: is repeated in Supp Info.

This method part has been removed from the Supplement

32) -251: with $H_{int0} = 0.05$ and some H_{int} measurements near 0.05, I believe that some of the intrinsic noise values may be overestimated by much more than 14%.

The referee is correct in that the technical error determined here is fairly high. When discussing this between the authors it became clear that the method that we adapted from Elowitz, 2002 is not adequate anymore, given that modern confocal microscopes work with quite narrow detection windows. In fact this forced us to perform the control with different settings than used for the experiments to enable the detection of GFP with the CFP channel. We therefore developed a new strategy to safely determine the technical errors: First, we determined the noise caused by the microscope by using a CHROMATEC slide. Using exactly the experimental setting for the YFP and the CFP channel we scanned the same region (without bleaching) 221 times. Thus the YFP/CFP ratios were constant in all images and any noise would come from the microscope. Second, we determined the handling error introduced by the selection of ROIs. Both types of mistakes were very minor and we therefore did not correct the data. The details of this analysis are now explained in the Supplement.

33) Fig1AB: Why are the resolution and quality of these images poorer than that of FigS1ABC?

The experiments shown in Fig1 study the fluorescence changes over time. We therefore used lower resolutions of 512*512 px to minimize bleaching effects. In all other cases we choose a resolution of 1024*1024 px.

34) Fig1 contains data from multiple leafs. Is the normalisation done on each leaf separately, or on the combined data set?

We did the normalization for each leaf separately as we consider it possible that noise may differ between leaves.

35) Fig3A: without the stomata cells data no large difference in extrinsic noise would be observed. Therefore it would be more convincing to show images of these cells too.

We are not sure whether we understand this comment correctly. Showing pictures depicting the noise in stomata cells is not possible as they are interspersed between epidermal cells on leaves. We therefore can only pick them out for a quantitative analysis.

36) Fig4: could the coupling be due to the recent common ancestry of the young leaf cells, instead of due to communication?

As sketched above, we addressed this question in more detail by analyzing the two-stage gene expression model, a linear model including mRNA and protein production. As we wanted to obtain a rough estimate how large the correlation induced by inheritance of mRNA and protein content could be, we employed a simple cell division model, avoiding all further complications stemming from cell growth. In our opinion, more detailed models would be not justified at the current stage of the research. According to our analysis, a simple inheritance of the mRNA and protein molecules would not be sufficient. However, the inheritance of cellular states/parameters (for example the number of ribosomes etc) that influence noise could be sufficient to explain our experimental values. This is now included in the main text and we discuss that both, inheritance of cellular states and mobility of relevant molecules can explain the co-variation in neighboring cells. Details about the analysis can be found in the Supplement.

37) SuppInfo 1.2: can the authors provide some intuition why minimizing function (1) estimates the degradation rate.

After the conversion of the green to the red proteins, one can describe the protein abundance by an exponential decay. Our strategy is to obtain an estimate for the degradation rate from each cell and then use the mean or the median for our further estimates instead of estimating the degradation rate from the mean of the data. Because the cost-function for the optimization involves exponentials, the optimal individual degradation rates cannot be calculated analytically, but one would need to refer to numerical methods. However, a logarithmic transformation converts the cost-function into a much simpler polynomial function, for which one can obtain the optimum analytically. For not too much noise the position of the optimum is only mildly affected by this transformation. We explain this now in the Supplement.

38)

FigS1-7: please let each figure be followed by its own legend.

Each supplementary figure is now followed by its own legend.

FigS3 legend: '100 m' -> '100 um', 'median=17.1' -> 'median=0.171'

this has been corrected, thanks.

39) FigS5C-F: does this plot show the bootstrap analysis? It looks more like a box plot of the noise in each quintiles. The distributions of the 1st and 5th quintiles in C and E look very similar, are the p values correct?

This confusion is caused by our mistake in that S5 and S6 were exchanged. We are sorry for that.

40)

- FigS6-7 legends: do not seem to correspond to their figures.

This confusion is caused by our mistake in that S5 and S6 are exchanged. We are sorry for that.

41) - 60: I don't believe that this technique where new protein production is quantified using a photoconvertible fluophore is well known. A reference to the paper that initialized the technique might be in place (Leung et al. Nat. Neurosci. 9, 1247-1256 (2006) or Raab-Graham et al. Science 314, 144-148 (2006)?)

We agree, this will help the readers to follow the argumentation. We now added the Leung paper as a reference.

Reviewer #3 (Remarks to the Author):

42)

My primary concern is the choice of promoters to study or perhaps it is a matter of better explaining the rationale for the choice. In my view, I think it would be enough to demonstrate the sources of noise in plant tissue and the underlying causes or consequences can be explored in future work. However, I thought the paper lacked adequate discussion in framing these results. The authors chose to examine the 35S cauliflower mosaic virus promoter. This is very strong and non-endogenous promoter. The rationale was as follows: "We chose this promoter as it is not obviously under the control of any plant specific pathway and should therefore enable us to study noise fairly independent from plant regulatory processes." I can't quite understand the rationale. Why would you want to study noise independently of plant regulatory processes? I guess this means independently of trans regulatory processes but that is typically the source of extrinsic noise. Also, that doesn't quite make sense either because presumably the 35S promoter is hijacking the plant's own trans regulatory mechanisms to initiate transcription? I don't understand the rationale. In fairness, the authors do a subset of experiments using the ubiquitin promoter. This is an endogenous promoter that is also expressed at high levels. I didn't see the rationale for choosing another regulatory region that mediates high expression.

We agree with the referee and now feel that the explanation was not very convincing. We now reduce the explanation for the choice of promoters to two aspects. First, we need fairly strong promoters to enable the detection of noise. Second, we used two different promoters to exclude that we are looking at a specific property of one promoter.

43) There may be a good reason for choosing highly expressed regulatory regions but it would also seem to warrant some discussion of the limitations of this approach. Could intrinsic noise be more prominent for more lowly and moderately expressed genes such that the author's conclusions are limited to promoters mediating high rates of transcriptions. One could image that promoter activity is dependent on the concentration of trans factors and that would have an

impact on noise? All this to say that it would be good to have more context for the rationale and results. In particular, the animal and yeast literature on this topic is quite extensive and it would have helped if the authors had drawn parallels for their choice of promoters, for example, to help us understand the rationale of the experiment and context for any conclusions. For example, I may be unaware of an aspect of that literature which really justifies the choice of reagents in this research and helps frame the result.

The reviewer is correct that intrinsic noise is expected to be highest for lowly expressed genes and it would be very interesting to be able to compare lowly to highly expressed genes *in planta*. However, our choice of the promoter is due to the fact that we need constitutive highly expressed promoters to obtain a measurable signal. Constitutive promoters are expected to reveal the lowest noise and therefore our measurements of the intrinsic noise are on the conservative side (Elowitz et al. 2002, Munsky 2012).

44)

Overall, I thought the authors explored a great question with an interesting approach but they needed more rationale of their experimental set up and a discussion of its implications.

In the current manuscript we have now extended the descriptions and discussions at various places to put the results into a biological context. In particular we add a discussion including theoretical analysis on the finding that noise is coupled between neighbouring cells.

Reviewers' comments:

Reviewer #1 (Remarks to the Author):

Schultheiß et al. responded in a positive and constructive way to most comments and improved the manuscript. However further clarification of some of the comments would be useful:

1. The authors analysed the distribution of CFP and YFP fluorescence intensities in Figure 2C and removed the samples displaying a skewing between CFP and YFP. However it would be better if the authors could still provide CFP and YFP fluorescence intensities distributions for the Figures 2, 3, S2, S3, S4 and S4 (analyses of extrinsic and intrinsic noise).

2. Regarding Figure S4, the authors conclude: "We found a subtle but significant correlation between nuclear size and extrinsic noise in mature leaves of p35S:2xNLS-YFP p35S:2xNLS-CFP and pUBQ10:2xNLS-YFP pUBQ10:2xNLS-CFP but no difference for intrinsic noise." The authors explained in their reply that as the method they are using is too crude to discuss differences between each quartile, they only comment differences between extreme quartiles. This is a reasonable argument but does not explain the fact that extrinsic variability for the 2nd quartile is much higher than for the 4th quartile for the pUBQ10:2xNLS-YFP pUBQ10:2xNLS-CFP plants.

Regarding this, it would be better if the authors could tone down their conclusion. Moreover, they could also provide CFP and YFP fluorescence intensities distributions for the different quartiles (see point 1). Indeed, it is difficult to see in the Figure S4D if extreme points (very low or high intensities) could explain the higher extrinsic noise observed for the 2nd and 3rd quartiles.

3. The authors improved the structure of the paper, extended descriptions and discussions. In this new structure, the 1st part of the result sections (temporal analysis of fluctuations) is now a little bit confusing. Perhaps the authors could describe better their model in a separate section. The model is then only used in the last section of the results where the authors analyse noise between neighbouring cells. Perhaps the authors could restructure the paper to make it easier for the reader to follow the link between the description of the model in the first section and its extension in the last section.

Minor comments:

P4 89-90: Could the authors explain better what they mean by "When comparing the fluorescence levels between the two time points we found correlations in the theoretically expected range"? Do they mean a correlation between 0.6 and 1 as explained in supplementary data?

P5 116-117: "Plotting the mean CFP values against the mean YFP values revealed intrinsic and extrinsic noise for young and old leaves (representative leaf shown in figure 2C)." The authors show a scatterplot of the mean CFP values against the mean YFP values for a young leaf only.

Reviewer #2 (Remarks to the Author):

I don't understand some of the changes in the new manuscript and it has not convinced me that it contains enough robust data for all of its claims.

The previous manuscript presented data for the fluorescence that showed skewing in mature leaves (Fig 2C). The rebuttal letter notes that some of this data is now removed ("We found that only very few samples displayed a skewing between CFP and YFP for unknown reasons. We therefore did not consider these samples in our analysis to ensure that our estimation of intrinsic and extrinsic noise is

not biased because of a potential skewing effect.”). I believe this new data treatment / selection is not described in the manuscript. The fluorescence data of mature leaves has now also been removed from the figure (F2C), which makes it impossible to judge how this new treatment affects the results and claims.

The extrinsic and intrinsic noise data presented in Fig2DE and Fig3BC have changed significantly since the previous version of the manuscript. From the rebuttal letter and text I understand that this is mainly due to different data treatment, and not due to additional experiments. In light of these changes, the presented data in Fig2E and Fig3C do not convince me that the intrinsic noise differs between tissues and cell types, as claimed in the abstract.

The first section presents data to support that protein expression fluctuates. In the previous version of the manuscript this was substantiated by showing that protein PRODUCTION in individual cells showed little correlation in subsequent time periods (Fig1CD). The revised manuscript has replaced these plots with data that shows that protein LEVELS are strongly correlated in subsequent time periods (Fig 1DE). I do not understand how figure 1DE contributes to the analysis of fluctuations in the expression. If there would be no fluctuations in expression, would one not expect to see a similar correlation graph, where deviations from perfect correlation are due to measurement noise?

The previous version of the manuscript stated that the measurement "error [may overestimate true] intrinsic noise values by a factor of 1.14 252 (14%)”. I made a minor comment about this estimation of 14%, because using the Elowitz formula together with values from the manuscript I found a maximum value close to 40%. In the current manuscript new measurements and a complex mathematical estimation of the errors are added. If I understand correctly, the error in measurement due to false pixel marking (e_m) is determined to be 6%, the error in measurement by the microscope (e_c and e_y) is lower than 1% and the error in determination intrinsic and extrinsic noise (f_{int} and f_{ext}) is 0.2%. With such small error for determination of extrinsic noise, doesn't this mean that the large variation in measured extrinsic noise (for example Fig2D young leave) is solely due to systematic differences in extrinsic noise between leaves (and not due to measurement error)? Either my understanding of the Supp Info is incorrect, or the measurement noise is highly underestimated.

As I already commented in my previous review, I feel that the first two sentences of the abstract (still) say opposite things: development is non-random despite molecular noise, development randomness may be due to molecular noise. I think what meant is more like “Although plant development is highly reproducible some stochasticity exists. This developmental stochasticity may be due to noisy gene expression.”.

Reviewer #3 (Remarks to the Author):

The manuscript "Stochastic gene expression in *Arabidopsis thaliana*" is more clear in revision and addresses some of my primary concerns.

One last comment on the issue of promoter choice. I revision of the rationale for choosing promoters makes more sense now. But that should lead to stronger caveats about any conclusions reached about intrinsic noise. Most genes are not expressed at the levels of either promoter used.

The conclusions about extrinsic noise and local domains are interesting and will contribute to the field of developmental biology.

I noticed reviewer #2 caught some important inconsistencies and seeming errors in analysis. In some

cases, those changes led to fairly dramatic revisions of the ms. Those seem to be corrected but I place a strong emphasis on making sure reviewer 2 is satisfied that the changes address those issues.

Overall, I think this is an important and interesting set of experiments that treads new ground in plant biology.

Response to Reviewers' comments:

Reviewer #1 (Remarks to the Author):

Schultheiß et al. responded in a positive and constructive way to most comments and improved the manuscript. However further clarification of some of the comments would be useful:

1. The authors analysed the distribution of CFP and YFP fluorescence intensities in Figure 2C and removed the samples displaying a skewing between CFP and YFP. However it would be better if the authors could still provide CFP and YFP fluorescence intensities distributions for the Figures 2, 3, S2, S3, S4 and S4 (analyses of extrinsic and intrinsic noise).

We have now added in our revised manuscript supplementary figures showing scatter plots for each sample and a cumulative scatter plot for each experiment. In addition, we used the Kolmogorov Smirnov test to ensure that the CFP and YFP values distributions are not significantly different in the biological replicas. The Kolmogorov Smirnov test reliably detects differences in distribution between CFP and YFP, be it location, scale, or family (seen as, e.g., skewing or tilted axis in the scatter plot) [REFERENCE: Handbook of Parametric and Nonparametric Statistical Procedures, David J. Sheskin, Taylor & Francis, FIFTH EDITION, CRC Press, 2011].

2. Regarding Figure S4, the authors conclude: "We found a subtle but significant correlation between nuclear size and extrinsic noise in mature leaves of p35S:2xNLS-YFP p35S:2xNLS-CFP and pUBQ10:2xNLS-YFP pUBQ10:2xNLS-CFP but no difference for intrinsic noise." The authors explained in their reply that as the method they are using is too crude to discuss differences between each quartile, they only comment differences between extreme quartiles. This is a reasonable argument but does not explain the fact that extrinsic variability for the 2nd quartile is much higher than for the 4th quartile for the pUBQ10:2xNLS-YFP pUBQ10:2xNLS-CFP plants. Regarding this, it would be better if the authors could tone down their conclusion. Moreover, they could also provide CFP and YFP fluorescence intensities distributions for the different quartiles (see point 1). Indeed, it is difficult to see in the Figure S4D if extreme points (very low or high intensities) could explain the higher extrinsic noise observed for the 2nd and 3rd quartiles.

We have now toned down the conclusions from this paragraph by saying that we observe a trend such that cells with higher endoreduplication levels have slightly more extrinsic noise.

In addition we provide four separate plots for each quartile instead of the cumulative plots shown in the previous figures S4A und S4D.

3. The authors improved the structure of the paper, extended descriptions and discussions. In this new structure, the 1st part of the result sections (temporal analysis of fluctuations) is now a little bit confusing. Perhaps the authors could describe better their model in a separate section. The model is then only used in the last section of the

results where the authors analyze noise between neighboring cells. Perhaps the authors could restructure the paper to make it easier for the reader to follow the link between the description of the model in the first section and its extension in the last section.

We have now restructured the paper and begin with the theoretical background, the model and the expectations. We hope that this structure makes it easier for the reader to follow the argumentations.

Minor comments:

P4 89-90: Could the authors explain better what they mean by “When comparing the fluorescence levels between the two time points we found correlations in the theoretically expected range”? Do they mean a correlation between 0.6 and 1 as explained in supplementary data?

Yes, this is correct. We now specify this in that particular sentence.

P5 116-117: “Plotting the mean CFP values against the mean YFP values revealed intrinsic and extrinsic noise for young and old leaves (representative leaf shown in figure 2C).” The authors show a scatterplot of the mean CFP values against the mean YFP values for a young leaf only.

We have now added a scatter plot of a mature leaf in the main figure and all individual scatter plots plus the cumulative scatter plot in the supplement.

Reviewer #2 (Remarks to the Author):

1. I don't understand some of the changes in the new manuscript and it has not convinced me that it contains enough robust data for all of its claims.

The previous manuscript presented data for the fluorescence that showed skewing in mature leaves (Fig 2C). The rebuttal letter notes that some of this data is now removed (“We found that only very few samples displayed a skewing between CFP and YFP for unknown reasons. We therefore did not consider these samples in our analysis to ensure that our estimation of intrinsic and extrinsic noise is not biased because of a potential skewing effect.”). I believe this new data treatment / selection is not described in the manuscript. The fluorescence data of mature leaves has now also been removed from the figure (F2C), which makes it impossible to judge how this new treatment affects the results and claims.

We now provide all scatter plots for each sample and the cumulative scatterplots of the YFP and CFP distributions. We also add the information in the material and method part that we excluded a few samples showing skewing.

2. The extrinsic and intrinsic noise data presented in Fig2DE and Fig3BC have changed significantly since the previous version of the manuscript. From the rebuttal letter and

text I understand that this is mainly due to different data treatment, and not due to additional experiments. In light of these changes, the presented data in Fig2E and Fig3C do not convince me that the intrinsic noise differs between tissues and cell types, as claimed in the abstract.

We apologize for the oversimplification in the abstract. For intrinsic noise we found a difference between intrinsic noise in young and mature leaves in two independent lines ($p=0.02$ and $p=0.005$, Wilcoxon rank-sum test). We now clearly specify this in the abstract by saying "Intrinsic noise differed between young and old leaves. Differences of extrinsic noise were found between several tissues and/or cell types".

3. The first section presents data to support that protein expression fluctuates. In the previous version of the manuscript this was substantiated by showing that protein PRODUCTION in individual cells showed little correlation in subsequent time periods (Fig1CD). The revised manuscript has replaced these plots with data that shows that protein LEVELS are strongly correlated in subsequent time periods (Fig 1DE). I do not understand how figure 1DE contributes to the analysis of fluctuations in the expression. If there would be no fluctuations in expression, would one not expect to see a similar correlation graph, where deviations from perfect correlation are due to measurement noise?

During the previous revision we corrected the description of the experiment as outlined in our last rebuttal letter. The referee is correct in that one would expect a perfect correlation in the absence of fluctuations. Thus any deviation from the perfect correlation is either due to measurement noise or fluctuation of the expression. As our measurement noise is low, the deviation seen in figure 1DE represents by en large the temporal fluctuation in the protein expression. In our theoretical analysis we defined the lower bound expected from the model for our experimental setup. The fluctuation of protein expression is documented by the difference to one (the perfect correlation), which is in the range of 0.6 and 0.8. In the current manuscript we also discuss what we learn about the temporal aspects of the fluctuations.

4. The previous version of the manuscript stated that the measurement "error [may overestimate true] intrinsic noise values by a factor of 1.14 252 (14%)". I made a minor comment about this estimation of 14%, because using the Elowitz formula together with values from the manuscript I found a maximum value close to 40%. In the current manuscript new measurements and a complex mathematical estimation of the errors are added. If I understand correctly, the error in measurement due to false pixel marking (e_m) is determined to be 6%, the error in measurement by the microscope (e_c and e_y) is lower than 1% and the error in determination intrinsic and extrinsic noise (f_{int} and f_{ext}) is 0.2%. With such small error for determination of extrinsic noise, doesn't this mean that the large variation in measured extrinsic noise (for example Fig2D young leave) is solely due to systematic differences in extrinsic noise between leaves (and not due to measurement error)? Either my understanding of the Supp Info is incorrect, or the measurement noise is highly underestimated.

Although it was only meant as a minor comment we took the argument of the referee

very serious as it potentially put the whole analysis into question. As it is not possible to use the previously applied strategy measuring GFP in both channels without changing the settings we very carefully determined the measurement noises at each step including microscopic noise, handling noise and the error in determining intrinsic and extrinsic noise as summarized by the referee. As the combined noise is low enough we trust that the fluctuation measurements are solid. Moreover, a systematic difference in extrinsic noise between leaves cannot be explained by these types of handling or technical measurement noise. Therefore, we think that extrinsic noise in tissues can vary between individual samples.

This could make sense. Even under our well-defined growth conditions it is well possible that individual leaves face different micro-environments such as slightly different light conditions at different places in the growth chamber or shading by other leaves, different stress caused by air-flow or pathogens below visible infection levels.

We noted the large variation of extrinsic noise between leaves before, however, felt that our current data are too preliminary to seriously discuss this aspect in the manuscript. To really make this point a complete own project is necessary to control and manipulate the conditions with a much higher sample number.

5. As I already commented in my previous review, I feel that the first two sentences of the abstract (still) say opposite things: development is non-random despite molecular noise, development randomness may be due to molecular noise. I think what meant is more like "Although plant development is highly reproducible some stochasticity exists. This developmental stochasticity may be due to noisy gene expression."

Thank you for your suggestion.

Reviewer #3 (Remarks to the Author):

1. The manuscript "Stochastic gene expression in Arabidopsis thaliana" is more clear in revision and addresses some of my primary concerns.

One last comment on the issue of promoter choice. I revision of the rationale for choosing promoters makes more sense now. But that should lead to stronger caveats about any conclusions reached about intrinsic noise. Most genes are not expressed at the levels of either promoter used.

We now add a sentence when introducing the promoters saying "Although this limits general conclusions, this procedure should result in a conservative estimation of intrinsic noise in our experiments as experimental data and theoretical considerations show that constitutive promoters show the lowest intrinsic noise".

2. The conclusions about extrinsic noise and local domains are interesting and will contribute to the field of developmental biology.

I noticed reviewer #2 caught some important inconsistencies and seeming errors in analysis. In some cases, those changes led to fairly dramatic revisions of the ms. Those seem to be corrected but I place a strong emphasis on making sure reviewer 2 is satisfied that the changes address those issues.

Overall, I think this is an important and interesting set of experiments that treads new ground in plant biology.

Thanks, we added more information to enable a better judgement and hope that we can address the concerns of reviewer 2 adequately.

Reviewers' comments:

Reviewer #1 (Remarks to the Author):

In general we are happy that the authors have attempted to make the requested changes. We are happy for the paper to be published, with the following corrections:

1. We still find the first section of the paper (The temporal analysis of fluctuation) confusing. It would be great if the authors could explain exactly what they are trying to test in this experiment, before describing the theoretical model. Also, the authors have added a description of the model, where it predicts autocorrelation between 0.6- 1, between the two timepoints. It would be useful if the authors could discuss the biological relevance, or put the number 0.6 into context, beyond stating that similar values are seen in human cells. In the human study cited, gene expression was measured for 50 – 100 h, at a resolution of 1 frame per 10 min. In this study, the authors look at 2 timepoints (3h and 6h). Is it OK to compare results from the two studies? Does using only 2 timepoints affect the accuracy of the autocorrelation estimation? It would be helpful if the authors could discuss these issues.
2. It would be useful for the authors to provide a construct list in the supplemental materials, which lists which figure the construct was used for, and how it was constructed (transient versus stable transformation).
3. Figure 1 has the same title as figure 2. It should be made clearer that 1B and 1C are simulations.
4. Line 190 – Is it possible to reword this sentence to make the meaning clearer? – ‘Thus, the covariance between two cells is for total correlation equal...’
5. Line 209 – ‘A distance dependent correlation of extrinsic noise was also found in hypocotyl and root tissues’ – Sorry if I missed something, is it worth pointing out that the root experiment was only done for the 35s promoter, and not for both promoters as in the rest of the paragraph?

Reviewer #2 (Remarks to the Author):

I want to restate that I believe that this manuscript contains important work. But I am not convinced that the data supports all claims robustly. I do believe that differences in intrinsic and extrinsic noise between different cell types / ages exist, and that this would be important for development. But I think there is a good chance that some of the differences that are found here have the opposite sign in reality than as claimed here.

The major issue I have is that I fear there is a problem with the quantification of noise. The large variation in extrinsic noise between samples of the same condition is possibly due to an unknown systematic measurement error, or developmental differences between samples (in which case I don't feel they fall within the extrinsic noise definition anymore). I agree with the author's response that it is also possible that individual samples experience different micro-environments. But if variation in experiment conditions can cause such large differences, I feel that strong claims about small differences (although seemingly significant) between conditions (eg. young and mature leaves) cannot be made. Claims that I feel are currently supported by the data are (I am leaving the coupling results out of consideration): extrinsic noise is larger than intrinsic noise in plants and extrinsic noise in stomata cells is lower than that of other cells. For support of the latter claim I do feel that images of stomata cells, and/or information about the mean expression levels in different tissues is required, as

lower signal-to-noise ratios could lead to overestimation of intrinsic noise and underestimation of extrinsic noise.

Some additional issues:

The authors have now added normalized data plots of the data underlying Figures 3 and 4 in the supplementary figures. It seems that the skewing between CFP and YFP signal in data from one leaf that was noted by reviewer #1 in a previous round seems to be more systematic, as it appears in some of the cumulative data plots. A visual analysis of Figures S6B, S8B, S9B, S11B and S15B suggests that its data does not distribute well around the X=Y line. Although the skewing may not be to dramatic, I am not sure what it can be caused by and whether that could affect the results.

Regarding the first section (Temporal analysis of fluctuations), I think that the modeling (Figure 1) is useful in the sense that one gets a better feeling of what processes underlie the experimental data in figure 2CD, but I feel it is used to extract more from this data than is realistic. The experimental data consists of differences in expression measurements at only 2 unique time points. Any offset from a linear correlation between these points is either due to measurement error or fluctuations in expression. With only 2 time points, quantification of auto-correlation times becomes very sensitive to measurement error. The latter is quantified indirectly using complex mathematics in the Supp Info. I do not feel this quantification and the data itself is of sufficient quality and quantity that the use of this model could lead to an accurate estimation of the fluctuation times.

I could not find an explicit description in the manuscript of how the experimental data is normalized (division by mean fluorescence of data set, or complete data set? normalization on variance?).

Although the manuscript lists which cell types have significant differences in extrinsic noise, I don't believe it actually mentions which one is larger than the other.

Figure 1 has an incorrect legend title. Most of the data in Figure1C seems irrelevant to the current study, i.e. only the data point at $t_2 - t_1 = 3$ is actually used for interpretation of experimental data. Modeling of extrinsic noise (CV) to be constant over time for individual cells has a problem: if auto-correlation for noise does not go to 0 doesn't it describe development instead of extrinsic noise?

Reviewer #1 (Remarks to the Author)

1. We still find the first section of the paper (The temporal analysis of fluctuation) confusing. It would be great if the authors could explain exactly what they are trying to test in this experiment, before describing the theoretical model.

The revised version initially states that we aim to detect a temporal correlation of gene expression in plant leaves. Next we describe the experimental setup in more detail to ensure that the rationale of the design is clear. In particular we try to convey that the chosen strategy was a result of optimizing the setup to enable the detection of differences between time points.

We then describe the experiments and point out that the autocorrelation values are below 1 and that this indicates the presence of fluctuations during these time intervals.

Next, we explain that we used mathematical modeling to test whether the linear two stage model is sufficient to explain the data and how the cell-to cell variability affects the decay of auto-correlation.

2. Also, the authors have added a description of the model, where it predicts autocorrelation between 0.6 - 1, between the two time points. It would be useful if the authors could discuss the biological relevance, or put the number 0.6 into context, beyond stating that similar values are seen in human cells. In the human study cited, gene expression was measured for 50–100 h, at a resolution of 1 frame per 10 min. In this study, the authors look at 2 time points (3h and 6h). Is it OK to compare results from the two studies? Does using only 2 time points affect the accuracy of the autocorrelation estimation? It would be helpful if the authors could discuss these issues.

We now explain why we included a mathematical analysis (see above) to evaluate our experimental data.

Because of the specific experimental setup and constraints (excised leaves, non-stationary system) we had to use ensemble averaging for calculating the autocorrelation function. For ergodic processes the ensemble averaging is equivalent to time averaging. Thus our strategy should in principle not affect the accuracy for the chosen time intervals. However, we agree with the referee that we can due to experimental constraints only obtain estimates for the autocorrelation and it may be better to refrain from making quantitative comparisons with results from other studies. We therefore only conclude that we can detect fluctuations in measured time intervals.

3. It would be useful for the authors to provide a construct list in the supplemental materials, which lists which figure the construct was used for, and how it was constructed (transient versus stable transformation).

We have added a new Table S1 summarizing this information

4. *Figure 1 has the same title as figure 2. It should be made clearer that 1B and 1C are simulations.*

This is now changed.

5. *Line 190 – Is it possible to reword this sentence to make the meaning clearer? – ‘Thus, the covariance between two cells is for total correlation equal...’*

We changed the text into:

“However, using the dual reporter p35S:2xNLS-YFP p35S:2xNLS-CFP plants for a cross-analysis (relating CFP to YFP and vice versa, see figure 5A) we can estimate the variance of the extrinsic noise (e.g. variability in ribosome number, transcription factor abundance^{23,24}) and the covariance between the extrinsic noise of neighbouring cells (figure 5B). The covariance between stochastically identical cells is equal to the variance of the extrinsic noise. Therefore, it is necessary to normalize the covariance using the variance of the extrinsic noise to obtain a measure between 1 and -1 for the correlation between neighbouring cells (Supplementary Information Sec. 3).”

6. *Line 209 – ‘A distance dependent correlation of extrinsic noise was also found in hypocotyl and root tissues’ – Sorry if I missed something, is it worth pointing out that the root experiment was only done for the 35s promoter, and not for both promoters as in the rest of the paragraph?*

We added this information to this sentence.

Reviewer #2 (Remarks to the Author):

I want to restate that I believe that this manuscript contains important work. But I am not convinced that the data supports all claims robustly. I do believe that differences in intrinsic and extrinsic noise between different cell types / ages exist, and that this would be important for development. But I think there is a good chance that some of the differences that are found here have the opposite sign in reality than as claimed here.

1. The major issue I have is that I fear there is a problem with the quantification of noise. The large variation in extrinsic noise between samples of the same condition is possibly due to an unknown systematic measurement error, or developmental differences between samples (in which case I don't feel they fall within the extrinsic noise definition anymore). I agree with the author's response that it is also possible that individual samples experience different micro-environments. But if variation in experiment conditions can cause such large differences, I feel that strong claims about small differences (although seemingly significant) between conditions (eg. young and mature leaves) cannot be made.

We have adopted the view of the referee and interpret now only clear and robust differences rather than relying merely on statistically differences of extrinsic or intrinsic noise. We now state that extrinsic noise is higher than intrinsic noise and that extrinsic

noise in stomata is lower than in most other tissues/cell types. The point that extrinsic noise can vary between tissues/cell types can still safely be made and we believe to be on the safe side with our interpretations. Along the same line we removed the statement on differences of intrinsic noise to avoid over-interpretations of small but significant differences.

We hope that this decision takes away some pressure on concerns of the exactness of the measurements, as we refrain from interpreting small differences. Nevertheless we like to make two comments on this:

First, the large variations between samples were found only in leaves and one would need to explain a systematic error only for this tissue. We kept the experimental procedure as constant as possible, all experiments were done by the same person, and we found no correlation of high or low values with particular experimental days. Thus by all means, we could not identify any additional source for noise other than those addressed by the controls.

Second, although we do not interpret noise differences between young and old leaves anymore, we like to note that we consider it very likely that there is large variation of extrinsic noise in leaves (which we did not note in other tissues). In this case, and in particular if this is true for some but not all tissues this would still fall in the definition of extrinsic noise. Internally, we call this phenomenon a second level of extrinsic noise – which we aim to determine in the future.

2. Claims that I feel are currently supported by the data are (I am leaving the coupling results out of consideration): extrinsic noise is larger than intrinsic noise in plants and extrinsic noise in stomata cells is lower than that of other cells. For support of the latter claim I do feel that images of stomata cells, and/or information about the mean expression levels in different tissues is required, as lower signal-to-noise ratios could lead to overestimation of intrinsic noise and underestimation of extrinsic noise.

We have toned down this paragraph as outline above to focus on the robust data. We have now added a figure (figure S26) showing the raw images of the different cell types and include the mean signal intensities in the corresponding figure legend. The intensity values of the stomata were in the same range (transgenic line 1) or only slightly higher (transgenic line 2) as compared to the other cell types/tissues. Only the young leaves exhibited consistently lower signal intensities. We have not added a discussion in the manuscript on this topic as we do not consider it an error that can be separated from the experimental background noise already analyzed.

3. The authors have now added normalized data plots of the data underlying Figures 3 and 4 in the supplementary figures. It seems that the skewing between CFP and YFP signal in data from one leaf that was noted by reviewer #1 in a previous round seems to be more systematic, as it appears in some of the cumulative data plots. A visual analysis of Figures S6B, S8B, S9B, S11B and S15B suggests that its data does not distribute well around the X=Y line. Although the skewing may not be too dramatic, I am not sure what it can be caused by and whether that could affect the results.

Because we find differences between individual instances of tissues, we base all our analysis on the CFP/YFP fluorescence measured in the individual instances (data sets). Therefore, the scatter plots shown in S5A, S6A, etc. are relevant and not the cumulative scatter plots shown in S5B, etc. The observed slight skewing or rather a tilt around 1 (due to the normalization) can be caused by constant, i.e., it does not vary from cell to cell, background fluorescence, which differs between CFP and YFP. Although we subtracted the background signal, there may still be a small background signal left due to measurement errors. The constant background signal, however, leads always to an *underestimation* of the intrinsic and extrinsic noise. In general, scatter plots may be misleading and a better and more robust way to analyze the data is a quantile-quantile (q-q) plot. A true tilt in the data will show up as well in a q-q plot. We analyzed all q-q-plots and could not detect a significant tilt. Furthermore, an even better measure is the Kolmogorov-Smirnov (K-S) test, which is as well based on the cumulative distributions and is robust even for small data sets. For all shown data the p-value of the K-S test was larger than 0.06 and for most of the data $p > 0.2$ holds. Based on the p-values of the K-S test we estimate the remaining background error in the data to be smaller than 10 % of mean CFP/YFP fluorescence.

4. Regarding the first section (Temporal analysis of fluctuations), I think that the modeling (Figure 1) is useful in the sense that one gets a better feeling of what processes underlie the experimental data in figure 2CD, but I feel it is used to extract more from this data than is realistic. The experimental data consists of differences in expression measurements at only 2 unique time points. Any offset from a linear correlation between these points is either due to measurement error or fluctuations in expression. With only 2 time points, quantification of auto-correlation times becomes very sensitive to measurement error. The latter is quantified indirectly using complex mathematics in the Supp Info. I do not feel this quantification and the data itself is of sufficient quality and quantity that the use of this model could lead to an accurate estimation of the fluctuation times.

It was not our intention to overuse the model and extract more from the data than is realistic. We introduce the model to provide a context for the measured correlation. We are well aware that we encounter many unknowns and this is why we give a (broad) range rather a more precise value for the expected autocorrelation. Calculation of the autocorrelation can be done for stationary processes either by time averaging or by ensemble averaging. However, for non-stationary processes the autocorrelation depends on two time points and a simple time averaging cannot be done. Ensemble averaging is (for ergodic processes) a valid procedure and is not more sensitive to

measurement errors than other analysis methods. We followed in the calculation of the autocorrelation the procedure used for analysis of the experimental data and because we do have no further information about the extrinsic noise, we used the lower bound of the autocorrelation to limit the range of the autocorrelation. Regarding the analysis of the measurement errors we very carefully analyzed all sources of error to our best knowledge. The mathematics may appear complex, but it is not as only simple algebra is used and all steps are laid out for the reader.

5. I could not find an explicit description in the manuscript of how the experimental data is normalized (division by mean fluorescence of data set, or complete data set? normalization on variance?).

The data was always normalized using the mean fluorescence of the data set. This is now mentioned in the Material and Method part.

6. Although the manuscript lists which cell types have significant differences in extrinsic noise, I don't believe it actually mentions which one is larger than the other.

In the current version we are not trying anymore to highlight the small but significant differences. Therefore it also does not appear to make sense to show the corresponding statistics anymore.

7. Figure 1 has an incorrect legend title. Most of the data in Figure1C seems irrelevant to the current study, i.e. only the data point at $t_2-t_1 = 3$ is actually used for interpretation of experimental data. Modeling of extrinsic noise (CV) to be constant over time for individual cells has a problem: if auto-correlation for noise does not go to 0 doesn't it describe development instead of extrinsic noise?

The figure title has been corrected. Further, we agree with the reviewer and truncated the plot. That the autocorrelation does not decay to zero is due to the approximation that the extrinsic noise is constant over time. This simplification is justified as the extrinsic noise, e.g. the number of ribosomes, vary on a longer timescale compared to the intrinsic noise. As we do have currently no detailed information about the correlation time of the extrinsic noise, this approximation can be considered as a zero-order approximation, which will capture the overall behaviour of the system.

REVIEWERS' COMMENTS:

Reviewer #1 (Remarks to the Author):

Ilka Araújo and colleagues have made a strong attempt to address the comments of both reviewers. The revised version of the manuscript is now easy to read and all claims appear well supported by the data. I have only a couple of minor points:

1. The authors state that the data and code are available on 'reasonable request'. I expect that there will be strong interest in the field in the data and constructs. Would it be possible for the reporter lines to be placed in a public repository, and the data made available in a public database?
2. Before publication, it would be great if the authors could add information in the legends about what the two colours in the scatterplots represent (grey and black), in figures 1 and 3.
3. P9 line 213, can the authors close the parenthesis and add a space: "p=0.0014; figure S21) and with a pUBQ10:2xNLS-YFP pUBQ10:2xNLS-CFP line" (instead of "p=0.0014; figure S21and with a pUBQ10:2xNLS-YFP pUBQ10:2xNLS-CFP line").